# TARGETED ADVERSARIAL SELF-SUPERVISED LEARNING

## ABSTRACT

Recently, unsupervised adversarial training (AT) has been extensively studied to attain robustness with the models trained upon unlabeled data. To this end, previous studies have applied existing supervised adversarial training techniques to self-supervised learning (SSL) frameworks. However, all have resorted to untargeted adversarial learning as obtaining targeted adversarial examples is unclear in the SSL setting lacking of label information. In this paper, we propose a novel targeted adversarial training method for the SSL frameworks, especially for positive-pairs in SSL framework. Specifically, we propose a target selection algorithm for the adversarial SSL frameworks; it is designed to select the most confusing sample for each given instance based on similarity and entropy, and perturb the given instance toward the selected target sample. Our method significantly enhances the robustness of a positive-only SSL model without requiring large batches of images or additional models, unlike existing works aimed at achieving the same goal. Moreover, our method is readily applicable to general SSL frameworks that only uses positive pairs. We validate our method on benchmark datasets, on which it obtains superior robust accuracies, outperforming existing unsupervised adversarial training methods.

## 1 INTRODUCTION

Enhancing the robustness of deep neural networks (DNN) is a critical challenge for their real-world applications. DNNs have been known to be vulnerable to adversarial attacks using imperceptible perturbations (Goodfellow et al., 2015), corrupted images (Hendrycks & Dietterich, 2019), and images with shifted distributions (Koh et al., 2021), which cause the attacked DNN models to perform incorrect predictions. A vast volume of prior studies has proposed to leverage adversarial training (AT) (Madry et al., 2018); AT explicitly uses generated adversarial examples with specific types of perturbations (e.g., $\ell_\infty$-norm attack) when training a DNN model. Most of these previous AT studies have considered supervised learning settings (Madry et al., 2018; Zhang et al., 2019; Wu et al., 2020; Wang et al., 2019) in which we can utilize class label information to generate adversarial examples. On the other hand, achieving robustness in a self-supervised learning (SSL) setting has been relatively understudied despite the recent success of SSL in a variety of tasks and domains.

SSL frameworks (Dosovitskiy et al., 2015; Zhang et al., 2016; Tian et al., 2020b; Chen et al., 2020; He et al., 2020; Grill et al., 2020; Chen & He, 2021) have been proposed to learn transferable visual representations by solving for pretext tasks constructed out of the training data (Dosovitskiy et al., 2015; Zhang et al., 2016). A popular SSL approach is contrastive learning (e.g., SimCLR (Chen et al., 2020), MoCo (He et al., 2020)), which learns to maximize the similarity across positive pairs, each of which contains differently augmented samples of the same instance, while minimizing the similarity across different intances. Recently, to establish robustness in these SSL frameworks, RoCL (Kim et al., 2020) and ACL (Jiang et al., 2020) have proposed adversarial SSL methods based on contrastive learning frameworks. They have demonstrated improved robustness without leveraging any labeled data. However, both of these adversarial SSL frameworks are inefficient as they require a large batch size in order to attain good performances either on clean or adversarial samples.

Recent SSL frameworks (Grill et al., 2020; Chen & He, 2021; Zbontar et al., 2021) mostly resort to maximizing the consistency across two differently augmented samples of the same instance, using an additional momentum encoder (Grill et al., 2020), without any negative pairs or additional

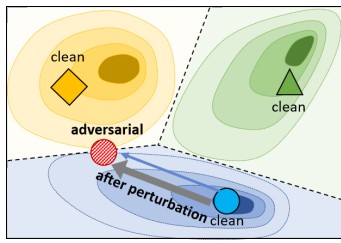 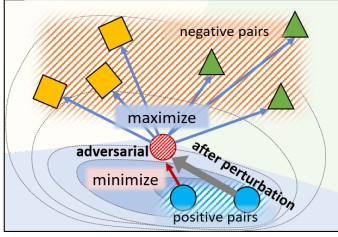 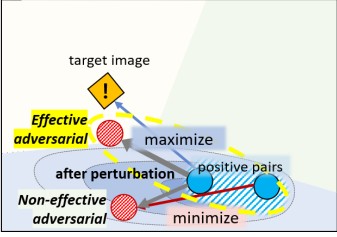

(a) supervised adversarial attack  (b) contrastive adversarial attack  (c) positive-only similarity adversarial attack

Figure 1: **Motivation.** In supervised adversarial learning (a), perturbation is generated to maximize the cross-entropy loss, which will push adversarial examples to the decision boundaries of other classes. In adversarial contrastive SSL (b), perturbation is generated to minimize the similarity (red line) between positive pairs while maximizing the similarity (blue lines) between negative pairs. Then, the adversarial examples may be pushed to the space of other classes as negative samples may mostly contain samples from other classes. However, in positive-only adversarial SSL (c), minimizing the similarity (red) between positive pairs have weaker constraints in generating effective adversarial examples than supervised AT or contrastive-based SSL. To overcome such a limitation, we suggest a selective targeted attack that maximizes the similarity (blue) to the most confusing target instance (yellow square in (c)).

networks (Chen & He, 2021; Zbontar et al., 2021). Such non-contrastive SSL frameworks using only positive pairs are shown to obtain representations with superior generalization performance compared to contrastive counterparts in a more efficient manner. However, leveraging untargeted adversarial attacks in these SSL frameworks results in a suboptimal performance. BYORL (Gowal et al., 2021a), an adversarial SSL framework using only positive pairs, obtains much lower robust accuracies than those of adversarial contrastive-learning SSL methods on the benchmark datasets (Table 3). Then, what is the cause of such suboptimal robustness in a non-contrastive adversarial SSL framework?

We observe that this limited robustness mainly comes from the suboptimality of untargeted attacks; adversarial examples generated by the deployed untargeted attacks are ineffective in improving robustness in non-contrastive adversarial SSL frameworks. As shown in Figure 1, the attack in the inner loop of the adversarial training loss, designed to minimize the distance between two differently augmented samples, perturbs a given example to a random position in the latent space. Thus, the generated adversarial samples have little impact on the final decision boundaries. Contrarily, in contrastive SSL frameworks, the samples are perturbed toward negative samples to maximize the instance classification loss, most of which belong to different classes. Thus, the ineffectiveness of the untargeted attacks in non-contrastive SSL frameworks mostly comes from their inconsideration of other instances.

To tackle this issue, we propose **T**argeted **A**ttack for **RO**bust **S**elf-**S**upervised learning (TAROSS). TAROSS is designed to enhance robustness of a non-contrastive SSL framework with only positive pairs, such as BYOL (Grill et al., 2020) and SimSiam (Chen & He, 2021), by conducting **targeted attacks**, which perturbs the given instance toward a target. However, this leads to the question of which direction we want to perform the targeted attack to, that is unclear in unsupervised adversarial learning without class labels. To address this point, we consider attacking the instance toward another instance, and further perform an empirical study of which target instances help enhance robustness as opposed to randomly selected target instances, in a targeted attack. Based on our observation, we propose a simple yet effective target selection algorithm based on the similarity and entropy between instances.

Our main contributions can be summarized as follows:

- We demonstrate that achieving comparable robustness in the positive-only self-supervised learning (SSL) with contrastive-based SSL is difficult due to ineffective adversarial inputs generated by untargeted attacks.

- We perform an empirical study of different targeted attacks for non-contrastive adversarial SSL frameworks using only positive pairs. Then, based on the observation, we propose a novel targeted adversarial attack scheme which perturbs the target sample in the direction to the most confusing instance to it, based on similarity and entropy.

- We experimentally confirm that the proposed targeted adversarial SSL framework is able to obtain significantly high robustness, outperforming the state-of-the-art contrastive- and positive-only adversarial SSL methods.

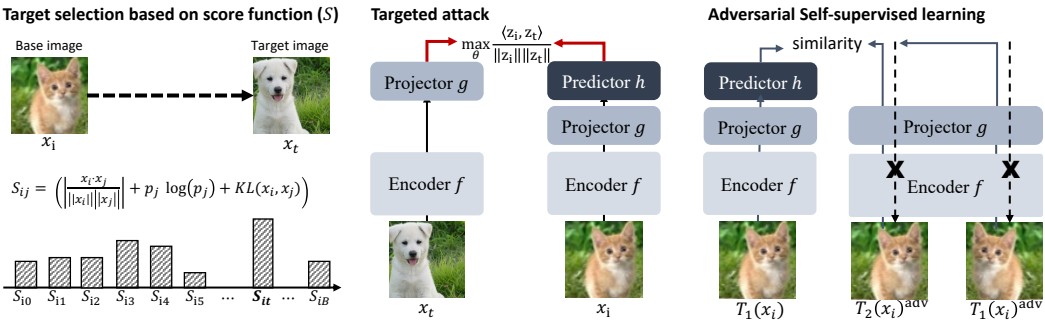

Figure 2: **Overview of TAROSS on SimSiam.** Our approach consists of three parts: target selection, targeted attack, and adversarial self-supervised learning (SSL). We propose a simple and effective similarity- and entropy-based target selection algorithm that selects the maximum score target based on score function ($\mathcal{S}$) (left). When we select the target image ($x_t$) for each instances, we conduct a targeted attack, which maximizes the similarity between the instance ($x_i$) and the targeted ($x_t$) (middle). Then, we train the model with targeted adversarial examples (right).

## 2    RELATED WORK

**Adversarial Training**    Szegedy et al. (2013) showed that imperceptible perturbation to an input image may lead a DNN model to misclassify a given input into a false label, demonstrating the vulnerabilities of DNN models against adversarial attacks. Goodfellow et al. (2015) proposed the Fast Gradient Sign Method (FGSM), which perturbs a given input to add imperceptible noise in the gradient direction of decreasing the loss of a target model. They also demonstrated that training a DNN model over perturbed as well as clean samples improves the robustness of the model against FGSM attacks. Follow-up works (Kurakin et al., 2016; Carlini & Wagner, 2017) proposed diverse gradient-based strong attacks, and Madry et al. (2018) proposed a projected gradient descent (PGD) attack and a robust training algorithm leveraging a minimax formulation; it finds an adversarial example that achieves a high loss while minimizing the adversarial loss over given data points. Due to a surge of interest in achieving robustness, various defense mechanisms (Song et al., 2017; Buckman et al., 2018; Dhillon et al., 2018) have been proposed.

However, Athalye et al. (2018) showed that many of the previous studies depend on gradient masking, which results in obfuscated gradients in the representation space. At the same time, this renders a target model highly vulnerable to stronger attacks that circumvent obfuscated gradients. TRADES (Zhang et al., 2019) proposed to minimize the Kullback-Leibler divergence (KLD) over clean examples and their adversarial counterparts, thus enforcing consistency between their predictions. They further showed that there is a theoretical trade-off relationship in achieving both clean accuracy and robustness. Recently, leveraging additional unlabeled data (Carmon et al., 2019) and conducting additional attacks (Wu et al., 2020) have been proposed. Carmon et al. (2019) proposed using Tiny ImageNet (Le & Yang, 2015) as pseudo labels, and Gowal et al. (2021b) proposed using generated images from generative models to learn richer representations with additional data.

**Self-Supervised Learning**    Due to the high annotation cost of labeling data, SSL has recently gained a large attention (Dosovitskiy et al., 2015; Zhang et al., 2016; Tian et al., 2020a;b; Zbontar et al., 2021). Previously, SSL focused on solving a pre-task problem of collaterally obtaining visual representation, such as solving the jigsaw puzzle (Noroozi & Favaro, 2016), predicting the relative position of two regions (Dosovitskiy et al., 2015), or impainting the masked area (Pathak et al., 2016). Contrastive SSL coined with data augmentation (Chen et al., 2020; He et al., 2020) has achieved impressive performance in SSL. On the other hand, other previous studies employed a momentum network to learn visual differences between two augmented images (Grill et al., 2020) or to mimic the momentum network with stop-gradient (Chen & He, 2021).

**Adversarial Self-Supervised Learning**    The first adversarial SSL method employed contrastive learning to achieve a high level of robustness (Kim et al., 2020; Jiang et al., 2020) without any class labels. Adversarial self-supervised contrastive learning (Kim et al., 2020; Jiang et al., 2020) generated an instance-wise adversarial example that maximizes the contrastive loss against their positive and negative samples by conducting untargeted attacks. Both methods achieved robustness

Table 1: Experimental results against PGD $\ell_\infty$ attacks on ResNet18 where all models are trained on the CIFAR-5 for simple observation. The CIFAR-5 includes airplane, automobile, bird, cat, and deer from CIFAR-10.

| Self-supervised framework | Method | Clean | PGD $\ell_\infty$ |
|---|---|---|---|
| *Contrastive based approach* | | | |
| SimCLR | ACL Jiang et al. (2020) | 80.84 | 39.16 |
| SimCLR | RoCL Kim et al. (2020) | 87.74 | 47.60 |
| *Non-contrastive based approach with untargeted attack* | | | |
| BYOL | Untargeted attack | 75.40 | 21.00 |
| SimSiam | Untargeted attack | 66.36 | 36.53 |
| *Non-contrastive based approach with targeted attack (**Ours**)* | | | |
| BYOL | Randomly targeted attack | 83.50 | **36.50** |
| BYOL | Entropy and Similarity based targeted attack | **87.08** | 35.18 |
| SimSiam | Randomly targeted attack | 77.08 | 47.58 |
| SimSiam | Entropy and Similarity based targeted attack | **79.54** | **49.50** |

with the cost of requiring large computation power due to a large batch size for contrastive learning. On the other hand, Gowal et al. (2021a) utilized only positive samples to obtain adversarial examples by maximizing the similarity loss between the latent vectors from the online and target networks, enabling this method free for batch size. However, it exhibited relatively poorer robustness than that of self-supervised contrastive learning, even with an advanced SSL framework. Despite this advanced framework, i.e., non-contrastive SSL, that employs only positive pairs, robustness is not guaranteed with a simple combination of untargeted adversarial learning and advanced SSL. To overcome such vulnerability in non-contrastive SSL, we propose a targeted attack leveraging a novel score function designed to improve robustness.

## 3    Targeted Adversarial Self-supervised Learning

Adversarial SSL has been proposed to apply existing supervised adversarial learning to models trained with SSL; this method generates adversarial examples in an untargeted manner and leverages these examples to improve robustness. Despite previous works being rather a straightforward combination of two approaches, we argue that leveraging untargeted adversarial attacks still leaves large room for robustness to improve.

### 3.1    Attack in Adversarial Self-supervised Learning

**Adversarial supervised training.**    To explain adversarial SSL, we first recap adversarial supervised training with our notations. We denote the dataset $\mathcal{D} = \{X, Y\}$, where $x \in X$ is a training example, and $y \in Y$ is its corresponding label. In this supervised learning task, a model is $f_\theta : X \to Y$, where $\theta$ is a set of the model parameters to train.

Given $\mathcal{D}$ and $f_\theta$, an *adversarial attack* generates an adversarial example by adding a perturbation to a given source image that maximizes the loss within a certain radius from it (e.g., $\ell_\infty$ norm balls). We define this adversarial $\ell_\infty$ attacks as follows:

$$\delta^{t+1} = \Pi_{B(0,\epsilon)}\Big(\delta^t + \alpha \mathtt{sign}\Big(\nabla_{\delta^t}\mathcal{L}_{\mathrm{CE}}\big(f(\theta, x + \delta^t), y\big)\Big)\Big), \tag{1}$$

where $B(0, \epsilon)$ is the $\ell_\infty$ norm-ball with radius $\epsilon$, $\Pi$ is the projection function to the norm-ball, $\alpha$ is the step size of the attacks, and $\mathtt{sign}(\cdot)$ is the sign of the vector. Also, the perturbation $\delta$ is the accumulated perturbations by $\alpha\mathtt{sign}(\cdot)$ over multiple iterations $t$, and $\mathcal{L}_{\mathrm{CE}}$ is the cross-entropy loss. In the case of PGD (Madry et al., 2018), the attack starts from a random point within the epsilon ball and performs $t$ gradient steps, to obtain a perturbed sample $x^{\mathtt{adv}}$.

*Adversarial training.* (AT) is a straightforward way to improve the robustness of the model by minimizing the training loss embedding the adversarial described above as an inner loop, as follows:

$$\mathcal{L}_{\mathrm{AT}} = \max_{\delta \in B(x,\epsilon)} \mathcal{L}_{\mathrm{CE}}\big(f(\theta, x + \delta), y\big). \tag{2}$$

**Untargeted adversarial self-supervised learning.**    Previously proposed adversarial SSL methods (Jiang et al., 2020; Kim et al., 2020; Gowal et al., 2021a) naturally combined the adversarial supervised training and SSL frameworks. Therefore, all previous works design the inner loop of

---

**Algorithm 1** Targeted Attack Robust Self-Supervised Learning (TAROSS)

---

**Require:** Dataset $\mathcal{D}$, transformation function $\mathbf{T}$, model $f$, parameter of model $\theta$, target score function $\mathcal{S}$, constant $w$

    **for** `iter` $\in$ number of iteration **do**
        **for** $x_i \in$ miniBatch $B = \{x_1, \dots, x_m\}$ **do**
            **for** `r` in 2 **do**
                transform input $\mathbf{T}_r(x_i)$
                Find target images $\mathbf{t}_r(x_j) = \mathcal{S}(\mathbf{T}_r(x_i), B)$
                Generate targeted adversarial examples $\mathbf{T}_r(x_i)^{adv} = \mathbf{T}_r(x_i) + \delta^t$
$$\delta^{t+1} = \Pi_{B(0,\epsilon)}\Big(\delta^t + \alpha \texttt{sign}\Big(\nabla_{\delta^t}\mathcal{L}_{\texttt{target}}\big(f(\theta, \mathbf{T}_r(x_i) + \delta^t), f(\theta, \mathbf{T}_r(x_j))\big)\Big)\Big)$$
            **end for**
            Calculate training loss
            $\mathcal{L}_{\texttt{TAROSS}} = \mathcal{L}_{\texttt{selfsup}}(\mathbf{T}_1(x_i), \mathbf{T}_1(x_i)^{adv}) + w \cdot \mathcal{L}_{\texttt{selfsup}}(\mathbf{T}_1(x_i)^{adv}, \mathbf{T}_2(x_i)^{adv})$
        **end for**
        Optimize the weight $\theta$ over $\mathcal{L}_{\texttt{TAROSS}}$
    **end for**

---

adversarial attack with untargeted attack as follows:

$$\delta^{t+1} = \Pi_{B(0,\epsilon)}\Big(\delta^t + \alpha \texttt{sign}\Big(\nabla_{\delta^t}\mathcal{L}_{\texttt{loss}}\big(f(\theta, \mathbf{T}_1(x) + \delta^t), f(\theta, \mathbf{T}_2(x))\big)\Big)\Big), \tag{3}$$

where perturbation is generated to maximize the self-supervised loss $\mathcal{L}_{\texttt{loss}}$ that minimizes the similarity between positive pairs, and maximize the similarity between negative pairs if negative pairs exists (Jiang et al., 2020; Kim et al., 2020).

**Targeted adversarial self-supervised learning**    We argue that leveraging untargeted adversarial attacks in an SSL framework using positive-only pairs still leaves large room for better robustness. We can simply improve robustness in non-contrastive SSL by changing the inner loop to a randomly targeted attack as follows:

$$\delta^{t+1} = \Pi_{B(0,\epsilon)}\Big(\delta^t + \alpha \texttt{sign}\Big(\nabla_{\delta^t}\mathcal{L}_{\texttt{target}}\big(f(\theta, x + \delta^t), f(\theta, x')\big)\Big)\Big), \tag{4}$$

where $\mathcal{L}_{\texttt{target}} = -\mathcal{L}_{\texttt{loss}}$, $x'$ is a randomly selected target within the batch. To precisely elaborate our observation in the latter section, we introduce a score function ($\mathcal{S}(x, \texttt{batch})$) to select the target ($x'$) within the batch. The score function ($\mathcal{S}$) can be a random-sampling function or a designed function as Eq. 6. In other words, the score function outputs the target ($x'$) corresponding to the base image ($x$), then the targeted attack generates perturbation that maximizes the similarity to target $x'$.

In Table 1, an untargeted attack makes the model more vulnerable when the non-contrastive SSL frameworks utilize only positive pairs, such as in BYOL (Grill et al., 2020), and SimSiam (Chen & He, 2021). However, when we modify the inner loop of the untargeted attack to the targeted attack with the simple random sampling score function as Eq. 4., non-contrastive SSL achieves better robustness.

## 3.2   Observations of Target Selection based on Score Function

In the previous section, we found that the targeted attack can improve robustness even with randomly selected target images in the batch. However, we argue there exist effective target images for each base input $x$ that can further improve robustness. However, selecting effective target instances for the targeted attack is not trivial. To enable this, we propose leveraging the K-means clustering-based and similarity-based score functions. We mainly observe our score function on SimSiam (Chen & He, 2021) and futher describe the details of our method for observations in Appendix C.

**K-means clustering-based target selection.**    Intuitively, perturbing the most confusing class can be a straightforward way of implementing a strong adversarial attack for better robustness. To validate our assumption, we first design the score function based on labels from K-means clustering. To find the most confusing instance, we label the pseudo class $y_i$ on the source images $x_i$ with K-means clustering and then find the most closest cluster $y_t$ to the pseudo class $y_i$ based on the centroid vector of each cluster. Accordingly, we can filter the instances that are labeled as $y_t$ from the batch.

Among those, the score function selects the instance that is closest to the source images $x_i$.

As shown in Table 2, the K-means clustering target selection score function shows better robustness than the random selection when performing targeted attacks. However, K-means clustering requires a lot of computation resources to derive a pseudo class label at every iteration, and it is vague to set the K value when there is no label information. Moreover, we cannot guarantee that the closest instance in the confused cluster is always the best candidate to improve robustness (Zhang et al., 2020) becase the closest cluster to an instance itself may differ from the closest cluster to the pseudo class cluster. For example, when the cat class cluster is close to the dog class cluster, there could be a cat instance close to horse images on the other side. Therefore, we also test the target selection score function based on the distance of each instance.

Table 2: Results of our TAROSS with K-means clustering based-, and similarity based- selection for targeted attack on CIFAR5.

| Selection | Clean | PGD $\ell_\infty$ |
|---|---|---|
| Untargeted* | 66.36 | 36.53 |
| Random | 77.08 | 47.58 |
| Similarity | 76.04 | 48.73 |
| K-means clustering | 75.86 | 49.39 |

**Similarity-based target selection.** In SSL, the similarity between given instances is an important metric to learn a good visual representation without any class label information (Chen et al., 2020; Tian et al., 2020b). Therefore, we leverage the similarity score to find optimal target images in the adversarial SSL. Before designing the score function based on the similarity score, we first observe the influence scores (Koh & Liang, 2017) of training points that are generated by the targeted attacks with randomly selected targets. The influence score is designed to approximate the influence function in DNN to compute the degree to which the training points are responsible for a given prediction (Koh & Liang, 2017). We followed Koh & Liang (2017) to calculate the influence scores of the adversarial examples that are generated by the targeted attack with a variety of target images. However, since this approximation requires a lot of computations, and the possible pair for the targeted attack is squared of the batch size, we sampled 60,000 training points to observe the correlation between the similarity and influence scores.

We found that the input pairs having the similarity of around $0.0$ have relatively high influence scores (Figure 7). To empirically verify whether the instances with the high influence score actually act as an effective target to show better robustness in adversarial SSL, we design an experiment in which we select target images based on the similarity. We experiment with five cases that have a different range of the similarity: r1: $-1.0 \sim -0.5$, r2: $-0.5 \sim -0.25$, r3: $-0.25 \sim 0.0$, r4: $0.0 \sim 0.25$, r5: $0.25 \sim 0.5$, and r6: $0.5 \sim 1.0$. As shown in Figure 3, when we select the target images between $-0.25 \sim 0.25$, it improves robustness compared to that using the random selection algorithm.

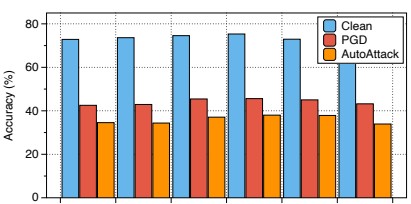

Figure 3: Robustness of the model that conduct targeted attack based on similarity score which selecting the target image that are in pre-defined range $(r1, \cdots, r6)$.

### 3.3 Targeted Attack with Entropy- and Similarity-based Score Function

In this section, we describe our overall framework to achieve robustness in non-contrastive SSL performing targeted attacks wherein targets are selected by leveraging our proposed score function. The target score function is designed based on the observation in the previous section.

**Positive-paired self-supervised learning.** Our targeted attack is designed for an SSL framework that only employs positive pairs of transformed images. Therefore, we mainly describe our method on the SimSiam (Chen & He, 2021).

We first describe an SSL method, SimSiam (Chen & He, 2021), that learns visual representations with only positive pairs using a stop-gradient. Let us denote the dataset $\mathcal{D} = \{X\}$, and transformation set $\mathbf{T}$ that augments the images $x \in X$. SimSiam consists of the encoder $f$, followed by the projector $g$, and the predictor $h$; each of $g$ and $h$ is a multi-layer perceptron (MLPs). To learn visual representations, SimSiam maximizes the cosine similarity between the positive pairs as follows:

$$\mathcal{L}_{\texttt{SimSiam}}(\mathbf{T}_1(x), \mathbf{T}_2(x)) = -\frac{1}{2}\frac{p_1}{||p_1||_2} \cdot \frac{z_2}{||z_2||_2} - \frac{1}{2}\frac{p_2}{||p_2||_2} \cdot \frac{z_1}{||z_1||_2}, \tag{5}$$

where $p_i = h(g(f(\mathbf{T}_i(x))))$ and $z_i = g(f(\mathbf{T}_i(x)))$ are output vectors of the predictor $h$ and the projector $g$, respectively, where $i$ stands for index of the differently augmented two images. Before calculating the loss, SimSiam stops the gradient on the $z$. Stop-gradient helps the model to form proper visual representations without any momentum network but makes the single model acts like a momentum network.

**Entropy and similarity based target selection.** From the observation in the previous paragraph, we argue that selecting target instances that are most confused and are adequately close to the base instance is effective for targeted attacks. For this, we design the score function based on the similarity and entropy values; it does not need any class information, as follows:

$$\mathcal{S}(\mathbf{T}_1(x_i), \mathbf{T}_2(x_j)) = \frac{e_i}{||e_i||_2} \cdot \frac{e_j}{||e_j||_2} + (p_j/\tau) \log (p_j/\tau), \tag{6}$$

where $p_j = h(g(f(\mathbf{T}_1(x_j))))$ and $e_i = f(\mathbf{T}_1(x_i))$ are output vectors of predictor $h$ and encoder $f$, respectively. Overall, the score function $\mathcal{S}$ consists of a cosine similarity term, and an entropy term. The cosine similarity is calculated between features of base images and candidate images in the differently augmented batch ($\mathbf{T}_2$). The entropy is calculated with the assumption that the vector $p$ is an instance's logit as Caron et al. (2021). We design the score function to work as a combination of selection algorithms based on K-means clustering and similarity values. It naturally selects the target that is mostly confused with base images (Figure 1). We also verify our score function selects targets as our intention in the experiment section (Figure 4b).

**Robust self-supervised learning with target attack (TAROSS).** In non-contrastive SSL, we make a positive pair, i.e., $\mathbf{T}_1(x), \mathbf{T}_2(x)$, with differently transformed augmentation. As shown in Figure 2, to generate adversarial examples, we first select the target images $\mathbf{T}_1(x)', \mathbf{T}_2(x)'$ for each base image $\mathbf{T}_1(x), \mathbf{T}_2(x)$, respectively, which have the maximum score from score function ($\mathcal{S}$). Here, we employ score function ($\mathcal{S}$) in Eq. 6. Then, we generate adversarial examples, i.e., $\mathbf{T}_1(x)^{adv}, \mathbf{T}_2(x)^{adv}$, for each transformed input with our suggested targeted attack (Eq. 4) where $\mathcal{L}_{\texttt{target}} = -\mathcal{L}_{\texttt{SimSiam}}$. Finally, we maximize the agreement between adversarial images $\mathbf{T}_1(x)^{adv}$, and $\mathbf{T}_2(x)^{adv}$ with clean image $\mathbf{T}_1(x)$ as follows:

$$\mathcal{L}_{\texttt{TAROSS}} = \mathcal{L}(\mathbf{T}_1(x), \mathbf{T}_1(x)^{adv}) + w \cdot \mathcal{L}(\mathbf{T}_1(x)^{adv}, \mathbf{T}_2(x)^{adv}) \tag{7}$$

where $\mathcal{L}$ is Eq. 5. Since all three instances have the same identity, we maximize the similarity between the clean image and adversarial examples. Overall, TAROSS is summarized in Algorithm 1.

## 4 EXPERIMENT

In this section, we validate our TAROSS on non-contrastive adversarial SSL frameworks (Section 4.1). Moreover, we evaluate the robustness of the representation with different datasets and in transfer-learning tasks (Section 4.2). Finally, we analyze why our targeted attacks help achieve better robustness compared to models using untargeted attacks (Section 4.3).

**Experimental setup.** We validate our TAROSS on top of BYOL (Grill et al., 2020) and Sim-Siam (Chen & He, 2021). To compare the effectiveness of TAROSS, we also implement non-contrastive adversarial SSL using untargeted attacks in each SSL framework. All the models are trained on ResNet18 with $\ell_\infty$ PGD attacks with an attack step of 10 with epsilon $8/255$. We evaluate the robustness against AutoAttack (Croce & Hein, 2020a) and $\ell_\infty$ PGD attacks with epsilon $8/255$ using an attack step of 20 iterations. We further describe the details of our experimental settings in Appendix B. The code will be available in Anonymous.

### 4.1 EVALUATING ROBUSTNESS

**Robustness with targeted/untargeted attack.** In Table 3, as we observed in the previous section, when replacing untargeted attacks with targeted attacks in positive-only SSL (BYOL, SimSiam), it obtains robustness comparable to contrastive adversarial SSL (RoCL, ACL), which verifies our motivation. Therefore, it is feasible to improve robustness even in positive-only SSL while attaining the pros of positive-only SSL frameworks. Interestingly, the targeted attacks helped improve the clean accuracy in SSL when compared with those using untargeted attacks in positive-only SSL. We conjecture that untargeted attacks are not only inadequate to learn robust features but also hinder to learn good visual representation of natural images.

Table 3: Experimental results against white box attacks on ResNet18 trained on the CIFAR10. We evaluate adversarial SSL with linear evaluation and robust linear evaluation. Clean is the accuracy of clean images. All models are evaluated with PGD $\ell_\infty$ with 20 steps of $\epsilon = 0.0314$ and AutoAttack (Croce & Hein, 2020a). Untargeted attack maximizes the training loss between the differently augmented but the same instances to generate adversaries. To see the effectiveness, we test TAROSS on positive-only SSL, i.e., BYOL, SimSiam.

| Train type | Self-supervised framework | Method | Clean | PGD $\ell_\infty$ | AutoAttack |
|---|---|---|---|---|---|
| Self-supervised linear evaluation | Contrastive | SimCLR RoCL (Kim et al., 2020) | 78.14 | **42.89** | 27.19 |
| | | SimCLR ACL (Jiang et al., 2020) | **79.96** | 39.37 | **35.97** |
| | Positive-only | BYOL Untargeted attack | 72.65 | 16.20 | 0.01 |
| | | BYOL **TAROSS** | **84.52** | 31.19 | 21.01 |
| | | SimSiam Untargeted attack | 71.78 | 37.28 | 32.41 |
| | | SimSiam **TAROSS** | 74.06 | **44.71** | **36.39** |
| Self-supervised robust linear evaluation | Contrastive | SimCLR RoCL (Kim et al., 2020) | 76.53 | **47.51** | 30.22 |
| | | SimCLR ACL (Jiang et al., 2020) | **77.17** | 40.67 | **39.13** |
| | Positive-only | BYOL Untargeted attack | 54.01 | 27.24 | 4.49 |
| | | BYOL **TAROSS** | 74.33 | 40.84 | 29.91 |
| | | SimSiam Untargeted attack | 68.88 | 37.84 | 31.44 |
| | | SimSiam **TAROSS** | **76.19** | **45.57** | **37.25** |

Table 4: Results of linear evaluation in STL10 and CIFAR10. Rob. stands for robust accuracy against PGD $\ell_\infty$ attack with 20 steps and $\epsilon = 0.0314$.

| Method | Attack | STL10 Clean | STL10 Rob. | CIFAR100 Clean | CIFAR100 Rob. |
|---|---|---|---|---|---|
| RoCL | Untarget | 52.63 | 19.19 | 45.99 | 17.17 |
| ACL | Untarget | 54.14 | 19.69 | 41.09 | 15.31 |
| BYOL | Untarget | **61.02** | 12.14 | **53.43** | 14.81 |
| SimSiam | Untarget | 43.88 | 12.70 | 27.53 | 14.20 |
| SimSiam | **TAROSS** | 46.38 | **21.32** | 36.02 | **22.18** |

Table 5: Results of adversarial transfer learning to CIFAR10 and STL10 from CIFAR100. Rob. stands for robust accuracy against $\ell_\infty$ PGD attack.

| CIFAR100 → Method | Attack | CIFAR10 Clean | CIFAR10 Rob. | STL10 Clean | STL10 Rob. |
|---|---|---|---|---|---|
| RoCL | Untarget | **73.93** | 18.62 | **74.06** | 19.06 |
| ACL | Untarget | 61.68 | 15.66 | 59.56 | 13.53 |
| BYOL | Untarget | 71.06 | 16.57 | 63.53 | 12.93 |
| SimSiam | Untarget | 50.05 | 25.43 | 34.54 | **24.82** |
| SimSiam | **TAROSS** | 50.50 | **25.44** | 43.13 | 22.46 |

**Robustness compared to contrastive-based approaches.** We compare our TAROSS to contrastive adversarial SSL methods to show that our approach could make positive-only SSL to achieve robustness comparable to previous works. To this end, we evaluate adversarial SSL with linear evaluation and robust linear evaluation against AutoAttack (Croce & Hein, 2020a) and PGD (Madry et al., 2018) $\ell_\infty$ attacks. Notably, as shown in Table 3, the targeted attacks allow the positive-only SSL model to have better robustness than that of contrastive adversarial SSL (RoCL, ACL) in linear evaluation. Specifically, when we train the fc layer, our model achieves 36.39% of robustness against strong AutoAttack which shows the effectiveness of our proposed method. Interestingly, the SimSiam-based model has already obtained good robustness in the linear evaluation compared to the robust linear evaluation which takes ×4 times to train. We conjecture that SimSiam-based TAROSS already has a robust well-cluster after the robust pretraining which does not need to find robust decision boundaries with the adversarial examples. Since the SimSiam directly maximizes the similarity between adversary and clean images only between the positive pairs with the single models, the gap between the representation of adversaries and clean examples may reduce. Moreover, our approach is also applicable to positive pairs in the contrastive-based approaches. As shown in Table 6, our TAROSS also could improve both clean and robustness of contrastive-based approaches. This verifies that targeted attack generates effective adversaries for self-supervised learning frameworks which is model agnostic approaches.

## 4.2 EVALUATING THE QUALITY OF ROBUST REPRESENTATION

**Robustness on multiple benchmarks datasets.** We validate our method on multiple benchmark datasets, such as CIFAR100 and STL10. In Table 4, our TAROSS consistently shows good robustness across different benchmark datasets when comparing with adversarial SSL frameworks using untargeted attacks. Especially, when the dataset becomes larger, such as CIFAR100, our method achieves even better robustness than contrastive learning-based approaches, e.g., RoCL, and ACL, where these approaches are sensitive to the number of same class samples in the single batch size. From this result, we argue that our targeted adversarial SSL contributes to learning better robustness in the larger dataset than contrastive-based SSL.

**Transfer to different data.** SSL is effective to utilize in several downstream tasks with a pretrained encoder. Therefore, we also evaluate how our robust pretrained features help in the different datasets in transfer tasks, which demonstrates the quality of our robust pretrained features in a different dataset.

We followed the experimental setting as supervised adversarial transfer learning tasks (Shafahi et al., 2020) which freeze the encoder and train only the fc layer. We pretrained the model on CIFAR100 and evaluate the robust transferability to STL10, and CIFAR10. Once our model is trained, we transfer the robust features to different kinds of datasets with fewer epochs and parameters, which is efficient in that we do not need to conduct adversarial training from scratch. In Table 5, our model also shows impressive transferability both in CIFAR10 and STL10 compare to untargeted adversarial SSL. Moreover, while our model shows relatively low clean accuracy, TAROSS could obtain about 10% gain in robustness compared to contrastive-based adversarial SSL. From these results, we confirm that our methods generate robust representations that can be transferred to several downstream tasks, which previous adversarial SSL studies addressed.

### 4.3 ANALYSIS OF TARGETED ATTACKS

In this section, we further analyze the targeted attacks in adversarial SSL to uncover evidence for why our proposed method is effective in adversarial SSL. We analyze which class is frequently selected when using our score function. Then, we visualize the difference between an untargeted attack and a targeted attack in the representation space.

**Analyze the distribution of the target class.**
We examine which class is selected as a target by the score function ($\mathcal{S}$). We test the selected target of the single class (airplane) to the adversarial supervised trained models (Madry et al., 2018) to obtain the probability and prediction for analyzing the class distribution. As shown in Figure 4b, half of the target images are the same class as the base image. And the next most frequently

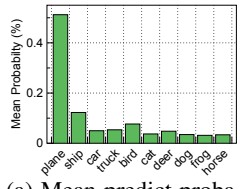 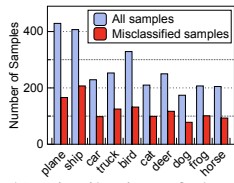

(a) Mean predict probability of base images

(b) Distribution of class of targeted images

Figure 4: Analysis of target from score function ($\mathcal{S}$)

selected class was the class with the second highest probability that the images of the corresponding class were most confused (Figure 4a). The score function we proposed selects images of the same class when the image in the center of the cluster, and images near the decision boundary that have a high probability to predict to a wrong class select images of a different class. Through these results, it can be confirmed that we designed a score function that appropriately uses K-means clustering to always select a different class from itself and heuristic target selection, which performs an instance-wise attack close to a similarity value, depending on the case, as we intended.

**Visualization of embedding space.** To observe the differences between images that are generated with targeted versus untargeted attacks, we visualize the embedding space of targeted attack examples and untargeted attack examples. In Figure 5, adversarial examples are denoted with dark blue, and clean examples are denoted with light blue; both are instances of the same class. As shown in Figure 5a, untargeted adversarial examples are

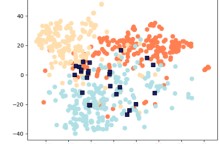 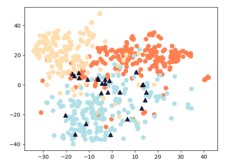

(a) Untargeted attack

(b) Targeted attack

Figure 5: Visualize embedding

located near clean examples. On the other hand, targeted adversarial examples are located near the boundary of the cluster (Figure 5b). This visualization shows that our targeted attack generates relatively more effective adversarial examples than untargeted attacks, which is likely to push the decision boundary to have a better robust representation.

## 5 CONCLUSION

In this paper, we showed that a simple combination of supervised adversarial training with SSL is highly suboptimal due to the ineffectiveness of adversarial examples generated by the untargeted adversarial attacks perturbed to random places without the consideration of decision boundaries. Based on this observation, we propose an instance-wise targeted attack scheme for an adversarial SSL framework. This scheme selects the target instance based on the similarity and entropy, such that the given instance is perturbed to be confused with the selected target. Our targeted adversarial SSL framework obtains representation that achieves better robustness than the state-of-the-art adversarial SSL frameworks, including contrastive ones, without using any negative pairs or additional networks. We believe that our work opens a door to future research on the search for more effective targeted attacks, for adversarial SSL.

## Reproducibility Statement

- **Datasets.** We use CIFAR10, CIFAR100, STL10, and CIFAR5 datasets for our experiment. To see more details, please see Supplementary B.2.
- **Baselines.** We use following models as our baseline: RoCL (Kim et al., 2020), ACL (Jiang et al., 2020), and BYORL (Gowal et al., 2021a). To see the training details, please see Supplementary A
- **Robustness Test.** We test our model and baselines against PGD attacks (Madry et al., 2018) and AutoAttack (Croce & Hein, 2020a). To see the evaluation details for robustness, please see Section 4. Experimental setup., and Supplementary B.4.
- **TAROSS.** To train our model, please see Supplementary B.3.
- **Table 1.** To reproduce the results in Table 1, please see Section 3.1.
- **Table 2.** To reproduce the results in Table 2, please see Section 3.2. K-means clustering-based target selection., and Supplementary C
- **Figure 3.** To reproduce the results in Figure 3, please see Section 3.2. Similarity-based target selection, and Supplementary C
- **Figure 4, and 5.** To reproduce results of Figure 4, and 5, please see Supplementary D.

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

# Supplementary Material

## Targeted Adversarial Self-Supervised Learning

## A  BASELINES.

- **RoCL (Kim et al., 2020).** RoCL is SimCLR (Chen et al., 2020) based adversarial self-supervised learning methods. We experiment with the official code[1]. To make a fair comparison, we set the attack step to 10 as other baselines. We train the model with 1,000 epochs under the LARS optimizer with weight decay $2e-6$ and momentum with 0.9. For the learning rate schedule, we also followed linear warmup with cosine decay scheduling. We set a batch size of 512 for all datasets (CIFAR10, CIFAR100, STL10). For data augmentation, we use a random crop with 0.08 to 1.0 size, horizontal flip with a probability of 0.5, color jitter with a probability of 0.8, and grayscale with a probability of 0.2 for RoCL training.

- **ACL (Jiang et al., 2020)** ACL is SimCLR (Chen et al., 2020) based adversarial self-supervised learning methods. We conduct the experiment with the official code[2]. To make a fair comparison, we set the attack step to 10 as other baselines. We train the model with 1,000 epochs. We set a batch size of 512 for STL10 dataset. For CIFAR10, and CIFAR100, we use the official pretrained checkpoints. For data augmentation, we use a random crop with 0.08 to 1.0 size, horizontal flip with a probability of 0.5, color jitter with a probability of 0.8, and grayscale with a probability of 0.2 for ACL training. We set PGD dual mode which calculates both clean and adversarial during the training.

- **BYORL (Gowal et al., 2021a)** BYORL is BYOL (Grill et al., 2020) based adversarial self-supervised learning methods for low label regime. Since there is no official code for BYORL we implement the BYORL by ourselves. Though we mentioned as BYOL with untargeted attack in the Table 1, 3, 4 and 5, please note that BYOL with untargeted approach stands for Gowal et al. (2021a). We implement based on BYOL from a self-supervised learning library[3]. We use the same CIFAR-10 setting in the library except for normalization. We exclude normalization in the data augmentation. To make a fair comparison, we implement on the ResNet18 with attack step 10 of PGD. As shown in supplementary materials in Gowal et al. (2021a), when the model is trained with 10 steps in ResNet34 it shows 37.88% of robustness. We conjecture that we have a different performance from the original paper because the original paper employs 40 steps of PGD in WideResNet34 to obtain the reported robustness which requires extraordinary computation power.

- **AdvCL (Fan et al., 2021)** AdvCL is SimCLR (Chen et al., 2020) based adversarial self-supervised learning which employ pseudo labels from the model that is pretrained on ImageNet (Krizhevsky et al., 2012) data. Even though the outstanding performance of AdvCL, we exclude this model as our baseline because the proposed methods require the model that is trained with the labels of ImageNet which we assume to have no label information for pretraining the model.

## B  DETAILED DESCRIPTION OF EXPERIMENTAL SETUPS

### B.1  RESOURCE DESCRIPTION.

All experiments are conducted with a two NVIDIA RTX 2080 Ti, except for the experiments with CIFAR100 experiments. For CIFAR100 experiments, two NVIDIA RTX 3080 are used. All experiments are processed in Intel(R) Xeon(R) Silver 4114 CPU @ 2.20GHz.

---

[1]https://github.com/Kim-Minseon/RoCL
[2]https://github.com/VITA-Group/Adversarial-Contrastive-Learning
[3]https://github.com/vturrisi/solo-learn

## B.2 Dataset description.

For experiments, we use CIFAR 10, CIFAR 100, CIFAR 5, and STL10. CIFAR 10 and CIFAR 100[4] consist of 50,000 training images and 10,000 test images with 10 and 100 classes, respectively. All CIFAR images are 32×32×3 resolution (width, height, and channel). CIFAR 5 is a subset of CIFAR 10 which contains all images from 5 classes: airplane, automobile, bird, cat, and deer. We also test with determined classes. The STL10[5] consists of 5,000 training images and 8,00 test images with 10 classes. All STL10 images are 96×96×3 resolution (width, height, and channel.) However, for our experiment in Table 4, 5, we resize the images into 32×32×3 resolution (width, height, and channel).

## B.3 Training detail.

For all methods, we train on the ResNet18 (He et al., 2016) with $\ell_\infty$ attacks with the attack strength of $\epsilon = 8/255$ and the step size of $\alpha = 2/255$, with the number of inner maximization iteration set to $K = 10$. For the optimization, we train every model for 800 epochs using the SGD optimizer with the learning rate of 0.05, weight decay of $5e-4$, and the momentum of 0.9. For data augmentation, we use a random crop with 0.08 to 1.0 size, horizontal flip with a probability of 0.5, color jitter with a probability of 0.8, and grayscale with a probability of 0.2. We exclude normalization for adversarial training. We set the weight of adversarial similarity loss $w$ as 2.0. We use batch size 512 with two GPUs.

In the score function, we calculate the similarity score term and entropy term as shown in Equation 6. First, to exclude the positive pairs' similarity score we set the similarity score between positive pairs to $-1$. Then, to calculate the overall score, after obtaining the similarity score and entropy of each sample, we normalize each component with Euclidean normalization to balance each component to score function.

## B.4 Evaluation details.

**PGD $\ell_\infty$ attack.** For all PGD $\ell_\infty$ attacks used in the test time, we use the projected gradient descent (PGD) attack with the strength of $\epsilon = 8/255$, with the step size of $\alpha = 8/2550$, and with the number of inner maximization iteration set to $K = 20$ with the random start.

**AutoAttack.** We further test against strong gradient based attack, i.e., AutoAttack (AA) Croce & Hein (2020a). AutoAttack is an ensemble attack of four different attacks (APGD-CE, APGD-T, FAB-T (Croce & Hein, 2020b), and Square (Andriushchenko et al., 2020)). AGPD-CE is an untargeted attack, APGD-T and FAB-T are targeted attacks. The Square is a black box attack. We use official code to test models[6].

**Self-supervised learning.** For self-supervised learning, we denote linear evaluation when we use only clean images to train the fully connected (fc) layer after the pretraining phase. When we denote robust linear evaluation, we train the fc layer with adversarial examples. While ACL uses partial fine-tuning to obtain their reported accuracy and robustness, to make a fair comparison we freeze the encoder and train only the fc layer. Robust fine-tuning is training all parameters including parameters of the encoder with adversarial examples. For linear evaluation, we followed baselines hyperparameters for each model. We train the baseline models with 150 epochs, 25 epochs, and 50 epochs for RoCL, ACL, and BYORL, respectively. We also followed their learning rate 0.1, 0.1, and $2 \times 10^{-3}$ for RoCL, ACL, and BYORL, respectively. On the other hand, we train our model with 100 epochs with a learning rate of 0.5 for linear evaluation. We use AT loss for robust linear evaluation except for ACL. For ACL, we use TRADES loss as the official code.

## C  Experiment Details of Our Targeted Attack

**K-means cluster-based TAROSS**  Intuitively, perturbing to the most confusing class is a straight-forward way of implementing a strong adversarial attack for better robustness. To validate our

---

[4] http://www.cs.toronto.edu/~kriz/cifar.html.
[5] https://ai.stanford.edu/~acoates/stl10/
[6] https://github.com/fra31/auto-attack

assumption, we design the score function based on labels from K-means clustering. We set K as 5 for CIFAR 5 in Table 2.

As show in Figure 6, we label the pseudo-class $y_i$ on the source images $x_i$ with the feature $z_i$. Then find the closest cluster $y_t$ to pseudo class $y_i$ based on the centroid vector $c$ of each cluster. We use the cosine similarity function to measure the distance between the latent vector $z$ and centroid vector $c$. To calculate the K-means cluster for each batch, K-means cluster-based TAROSS takes high computation.

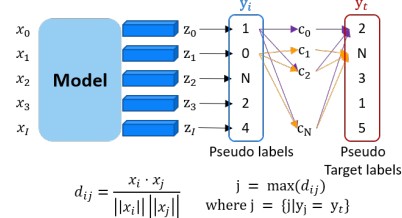

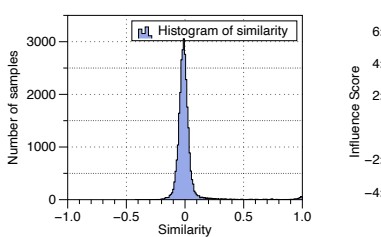

Figure 6: K-means clustering target selection

**Similarity score based TAROSS**
To calculate the influence score during the training, we employ underfitted adversarial self-supervised model which learns visual representation a little with 400 epochs. To calculate all pairs of the training set, it requires $\mathbf{O}(\mathcal{D}^2)$. Moreover, the computation of influence score (Koh & Liang, 2017) for single pair also takes a lot due to approximation. Therefore, we randomly sampled a subset of the dataset from the training set. We use official code of influence score[7]. As shown in Figure 3, samples that are in $-0.25 \sim 0.25$ shows relatively high influence score.

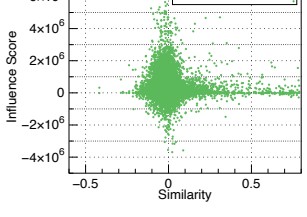

(a) Histogram of cosine similarity between sampled pairs.

(b) Scatter of influence score based on similarity

Figure 7: Analysis with influence score.

## D    EXPERIMENTAL DETAILS OF ANALYSIS

**Analysis the distribution of target class**    To analyze the target from the score function ($\mathcal{S}$), we employ an adversarially supervised trained model. We calculate the score function that is trained with our TAROSS on SimSiam. We use a train set. For each class, we calculate the mean predict probability which is the average of all softmax outputs of target images from the supervised trained model. Further, we also count the number of samples that are predicted for each class. In Figure 4, the results are target images of the airplane as a base image. There is a similar tendency even though we change the base class to other classes as shown in the following Figure 8.

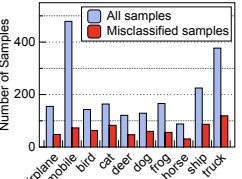
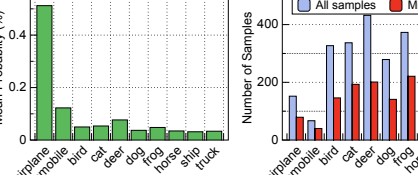
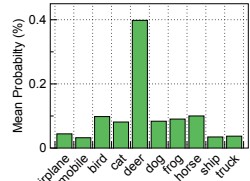

(a) Distribution of class of target of *automobile*.

(b) Mean predict probability of *automobile*

(c) Distribution of class of target of *deer*.

(d) Mean predict probability of *deer*

Figure 8: Analysis of target distribution in different class

**Visualization of embedding space**    To visualize the embedding of our targeted attack and untargeted attack, we use t-Distributed Stochastic Neighbor Embedding (t-SNE) (Chan et al., 2019) with the cosine similarity metric. Our TAROSS model is trained on CIFAR10 as a feature extractor. We sample a few examples and conduct two types of attacks, untargeted attack, and targeted attack. To visualize more effectively we ignore the other seven classes in CIFAR10. We visualize clean examples from three classes and then visualize adversaries that are generated with our targeted attack and untargeted attack, respectively with dark blue.

---

[7]https://github.com/kohpangwei/influence-release

# E ADDITIONAL EXPERIMENT

## E.1 CONTRASTIVE BASED ADVERSARIAL SELF-SUPERVISED LEARNING WITH TAROSS.

Our TAROSS could be also applied to positive pairs in contrastive-based adversarial self-supervised learning (e.g., RoCL (Kim et al., 2020), ACL (Jiang et al., 2020)). We applied our TAROSS in instance-wise attack of the contrastive-based approaches as follow,

$$\mathcal{L}_{\texttt{attack}} = \mathcal{L}_{\texttt{nt-xent}}(x, \{x_{\texttt{pos}}\}, \{x_{\texttt{neg}}\}) + \mathcal{L}_{\texttt{similarity}}(x, \{x_{j_{\texttt{TAROSS}}}\}) \tag{8}$$

where attack loss is consists of original attack loss nt-xent loss (Chen et al., 2020) and similarity loss. The similarity loss additionally constrains the positive pairs as the TAROSS that maximize the similarity between the $x$ with the $j^{th}$ index images which is searched by our TAROSS score function. Overall, we generate adversarial examples that maximizes the $\mathcal{L}_{\texttt{attack}}$ loss. Surprisingly, when we apply TAROSS on the contrastive learning based approach, previous work could achieve marginally better clean accuracy and robustness (Table 6). This shows that our empirical assumption also holds on contrastive-based SSL but since there is (1/batch size) effects on the total loss the gain could be marginal.

Table 6: Results of contrastive learning approach with TAROSS.

| | Linear evaluation | | Robust linear evaluation | |
|---|---|---|---|---|
| | Clean | PGD | Clean | PGD |
| RoCL | 78.14 | 42.89 | 76.53 | **47.51** |
| RoCL +TAROSS | **78.43** | **43.91** | **80.11** | 46.47 |
| ACL | 79.96 | 39.37 | 77.17 | 40.67 |
| ACL +TAROSS | **80.02** | **40.18** | **79.84** | **42.29** |

## E.2 ABLATION EXPERIMENT ON THE SCORE FUNCTION

Our score function consists of two terms: entropy term and cosine similarity term. While our score function is designed based on our observation in Section 3.2, we also empirically validate each term with the ablation experiment. We only use each term as the score function during the adversarial training as follow,

Table 7: Results of ablation study on score function.

| | Clean | PGD | AutoAttack |
|---|---|---|---|
| $\mathcal{S}_{\texttt{entropy}}$ | 78.43 | 40.35 | 32.51 |
| $\mathcal{S}_{\texttt{similarity}}$ | 72.90 | 44.59 | 36.12 |
| $\mathcal{S}_{\texttt{TAROSS}}$ | 74.06 | 44.71 | 36.39 |

$$\mathcal{S}_{\texttt{entropy}}(\mathbf{T}_1(x_i), \mathbf{T}_2(x_j)) = (p_j/\tau) \log (p_j/\tau), \tag{9}$$

$$\mathcal{S}_{\texttt{similarity}}(\mathbf{T}_1(x_i), \mathbf{T}_2(x_j)) = \frac{e_i}{||e_i||_2} \cdot \frac{e_j}{||e_j||_2}. \tag{10}$$

Through the experimental results in Table 7, the entropy term leverage to have good clean accuracy while similarity term focus on to have better robust performance. With combined score function, our model could have good robustness while having good clean accuracy.

## E.3 ROBUSTNESS AGAINST BLACK BOX ATTACK

We conduct black box attack to verify our model is robust to gradient free attacks. We generate black box adversaries with AT (Madry et al., 2018) model, RoCL (Kim et al., 2020) model and our models. Then, we test adversaries to each other. As show in the table, our model is able to defend the black box attack from AT model than the RoCL model. Moreover, our model generates stronger black box adversaries than RoCL since AT model shows more weak robustness.

Table 8: Results of black box attack. Models on the row are the tested models. Models on the columns are the source models to generate black box adversaries.

| | AT | RoCL | Ours |
|---|---|---|---|
| AT | - | 59.73 | **60.92** |
| RoCL | 70.40 | - | 57.98 |
| Ours | **69.97** | 54.99 | - |

