# OpenReview forum: "Targeted Adversarial Self-Supervised Learning"
_ICLR.cc/2023/Conference — Submitted to ICLR 2023_

### Official Review · Reviewer_yLsz · 2022-10-16

**Confidence:** 3
**Correctness:** 3
**Technical Novelty And Significance:** 2
**Empirical Novelty And Significance:** 2
**Recommendation:** 6

**Clarity, Quality, Novelty And Reproducibility:**

Clarity: the abstract is confusing. Other parts are clear.

Quality: good.

Novelty: the intuition is interesting.

Reproducibility: the intuition looks reasonable to me. It says " The code will be available in Anonymous" but seems there is no link. The other experiment results from other Github repositories (e.g., RoCL) looks good to me.

**Strength And Weaknesses:**

Strength:

The paper explains the intuition and numerical experiments clearly.

Weakness:

[1] From the intuition, it seems that the idea behind the proposed method is similar to contrastive learning. Both of them consider separate samples of different properties (e.g, class, or similarity). Could the authors provide some more insights on the difference between the proposed algorithm and the idea behind contrastive learning? Is there any interesting novel thoughts in the new method?

[2] Is there any specific reason on why this paper considers improving non-contrastive SSL? The numerical performance of the proposed algorithm does not outperform compared to contrastive learning. In all Table 3, 4, 5, the new method may either have a similar (or slightly worse) performance compared to contrastive based approach, or has a better robustness while sacrificing the clean testing performance (i.e., clean and adversarial trade-off).

The authors need to provide some explanation why they want to compare the proposed method only with non-contrastive SSL, but nor contrastive-SSL. For example, the authors may want to explain whether there is any concern to utilize contrastive learning. Or is there any particular area of study which cannot use contrastive learning? Or is there any potential advantage of using non-contrastive SSL? Or can the proposed method be applied in contrastive learning?

It is acceptable to not consider non-contrastive SSL for a theory paper. But since this paper aims to propose a new method, it is essential to demonstrate that the proposed method is good enough. There may be some scenarios that non-contrastive SSL is used instead of contrastive SSL, but it is not clarified in this paper. In all the experiments in this paper, it seems that we can always use contrastive SSL and get a better result.

Please also emphasize that the paper works on non-contrastive SSL in the abstract. My first impression was that this paper studies contrastive learning with targeted SSL. It is slightly confusing.

[3] Similarly, the observation in Figure 5 is not apperant.

[4] In the tables, the bold numbers distract readers from catching the groups with the best performance. Please adjust the tables to make them clear.


**Summary Of The Paper:**

This paper proposes a targeted adversarial training method for non-contrastive self-supervised learning (SSL) to improve the performance. Using the proposed algorithm, non-contrastive SSL gets an improvement in the robustness of the downstream tasks.

**Summary Of The Review:**

The main concern is to provide some illustration on why we want to consider the proposed algorithm. From all the numerical experiments in the current presentation of the paper, it seems that contrastive-SSL is a good method. The authors may consider adapting the proposed method into contrastive learning to further improve the performance, or provide some motivation why we some times can only use non-contrastive learning.

---

> ### Author Response · Authors · 2022-11-12
> **Response to comments (2/2)**
>
> **3. Similarly, the observation in Figure 5 is not apparent.**
> - Sorry for the unclear description in Figure 5. We clarify in the supplementary page 15.
> - We use tSNE for visualizing the features of clean samples and untargeted samples.
>
> **4. In the tables, the bold numbers distract readers from catching the groups with the best performance. Please adjust the tables to make them clear.**
> - Sorry for confusion. As you commented, we clarify the best performance in the table with the bold.
> - However, please note that our main comparison is untargeted adversarial self supervised training of positive-only based self-supervised learning frameworks [Gowal et al.].
>
> [Gowal et al.] Self- supervised adversarial robustness for the low-label, high-data regime, ICLR 2021

---

> ### Author Response · Authors · 2022-11-12
> **Response to comments (1/2)**
>
> **1. [Difference between proposed methods and contrastive learning]**
> > From the intuition, it seems that the idea behind the proposed method is similar to contrastive learning. Both of them consider separate samples of different properties (e.g, class, or similarity). Could the authors provide some more insights on the difference between the proposed algorithm and the idea behind contrastive learning? Are there any interesting novel thoughts in the new method?
> - We would like to emphasize that our approach is completely different from contrastive learning. The contrastive learning considers different samples as negative pairs to learn similarity differences while augmented samples are positive pairs because they have the same identity. However, we only focus on how to obtain robustness **without any negative pairs in adversarial non-contrastive SSL which is a non-trivial problem.**
> - In our approach **we try to find a useful target for targeted attack to generate effective adversaries** in adversarial self-supervised learning. In the non-contrastive SSL, positive pairs were enough to learn good visual representation in clean image training. However, we first found positive pairs could not generate effective adversaries in adversarial non-contrastive SSL.
> - To overcome such vulnerability, **we empirically observe diverse selection to generate effective adversaries in Section 3.** Interestingly, we found that **selecting confused instances as target for targeted attack could reach better robustness** while previous untargeted adversarial self-supervised learning could not.
>
> **2. [Why do we have to improve non-contrastive SSL?]**
> > Is there any specific reason on why this paper considers improving non-contrastive SSL? The numerical performance of the proposed algorithm does not outperform compared to contrastive learning. In all Table 3, 4, 5, the new method may either have a similar (or slightly worse) performance compared to contrastive based approach, or has a better robustness while sacrificing the clean testing performance (i.e., clean and adversarial trade-off). \
> > The authors need to provide some explanation why they want to compare the proposed method only with non-contrastive SSL, but nor contrastive-SSL. For example, the authors may want to explain whether there is any concern to utilize contrastive learning. Or is there any particular area of study which cannot use contrastive learning? Or is there any potential advantage of using non-contrastive SSL? Or can the proposed method be applied in contrastive learning? [$\cdots$]
>
> - We believe **when the computation budgets are limited such as in the edge devices, batch-free non-contrastive SSL could be applicable** than contrastive SSL which needs large batch-size.
> - Moreover, since **recent state-of-the art self-supervised learning models are developed to employ only positive pairs** [Grill et al.,Chen et al, Zbontar et al.], we believe **it is necessary to explore the robustness in non-contrastive SSL.**
> - Furthermore, **our approach is also applicable to positive pairs in contrastive SSL** as shown in following table. Surprisingly, our approach also could improve both clean and robustness in contrastive-SSL framework. This demonstrates selecting good positive pairs with targeted attack contributes a lot to construct good robust representation no matter what the self-supervised learning framework is.
>
> |   Linear evaluation          | Clean     |    PGD     |
> |-----------|:-----------:|:---------:|
> | RoCL | 78.14     |   42.89   |
> | RoCL **+TAROSS** |  **78.43**   | **43.91**    |
> | ACL | 79.96 | 39.37 |
> | ACL **+TAROSS** | **80.02**| **40.18** |
>
> |   Robust linear evaluation         | Clean     |    PGD     |
> |-----------|:-----------:|:---------:|
> | RoCL | 76.53 | **47.51**  |
> | RoCL **+TAROSS** |  **80.11** | 46.47  |
> | ACL | 77.17 | 40.67 |
> | ACL **+TAROSS** |**79.84** | **42.29** |
>
> - Especially, in table 1, and 3, 4, our work with non-contrastive SSL **shows better robustness against PGD attack and strong AutoAttack** than contrastive SSL. Moreover, as we expected, our approach with non-contrastive SSL demonstrates better transferred robustness which indicates our model benefits the high visual quality of non-contrastive SSL while contrastive SSL could not.
> - Therefore, we believe to have better robustness regardless of batch size, **a non-contrastive SSL approach is a better choice** than contrastive SSL. However, as you commented, when the scenarios does not have limitation on the budget, **contrastive SSL with our approach also could give better performance** than previous work.
>
> [Grill et al.] Bootstrap your own latent: A new approach to self-supervised Learning, arXiv 2020 \
> [Chen et al.] Exploring Simple Siamese Representation Learning, CVPR 2021 \
> [Zbontar et al.] Barlow Twins: Self-Supervised Learning via Redundancy Reduction, ICML 2021

---

> > ### Author Response · Authors · 2022-11-18
> > **Additional response on the "Why do we have to improve non-contrastive SSL?"**
> >
> > - The self-supervised framework has been developed recently that **does not limit the batch size and is able to obtain good performance** with only positive pairs [2,3,4] while **vulnerability has never been considered.**
> > - We believe our first observation that a simple combination of adversarial learning and recent positive-only self-supervised could not have enough robustness is a surprising and important observation. Furthermore, we adequately demonstrate our intuition and the methods to overcome such vulnerability.
> > - Moreover, we believe that not only for state-of-the-art performance, but also for **discovering under explored problem and suggesting appropriate solutions based on the problem is also meaningful** in this community. Furthermore, we believe that **reverting recent SSL frameworks to contrastive learning does not seem a genuine solution** to ensuring the robustness of non-contrastive self-supervised learning frameworks.
> > - We understand our approach could not show superior robustness in every task compared to contrastive-based learning, but **we resolve the vulnerability problem that non-contrastive SSL has where our work makes them have sometimes better robustness** than contrastive-based approaches.
> > - In table 1, and 3, 4, our work with non-contrastive SSL shows **better robustness against PGD attack and strong AutoAttack** than contrastive SSL. Moreover, as we expected, our approach with non-contrastive SSL demonstrates **better transferred robustness** which indicates our model benefits the high visual quality of non-contrastive SSL while contrastive SSL could not.
> >
> > [1] Gowal et al., Self- supervised adversarial robustness for the low-label, high-data regime, ICLR 2021 \
> > [2] Grill et al. Bootstrap your own latent: A new approach to self-supervised Learning, arXiv 2020  \
> > [3] Chen et al. Exploring Simple Siamese Representation Learning, CVPR 2021 \
> >  [4] Zbontar et al. Barlow Twins: Self-Supervised Learning via Redundancy Reduction, ICML 2021

---

> > > ### Comment · Reviewer_yLsz · 2022-11-21
> > > **Thanks for your response**
> > >
> > > I have read the review of other reviewers and the author responses. I think the authors need to improve the introduction and experiment sections to address the concerns. In particular,
> > >
> > > [1] The authors may consider adding paragraphs in the introduction to emphasize why sometimes contrastive learning is not preferred. I think the argument that contrastive learning is slow is a good direction.
> > >
> > > [2] I also suggest the authors to upload the running time of each algorithm to verify that contrastive learning is slow. In the current presentation, we only know the accuracy difference between contrastive learning and other algorithms, but not the other metrics, e.g. time.
> > >
> > > I'm considering raising my score, please update the manuscript based on these two comments if possible. If the manuscript cannot be updated now, could you reply me with the new introduction paragraphs and the experiment details? Thanks.

---

> > > > ### Author Response · Authors · 2022-11-25
> > > > **Thank you for your valuable comments**
> > > >
> > > > Thank you for your quick reply.
> > > >
> > > > First of all, contrastive learning itself is not a bottleneck of the training time in adversarial SSL since **an adversarial attack, which is iterative, requires larger computations than the calculation of the contrastive loss**. ACL and ours generate two adversarial examples to train the models while RoCL generates only one adversary which costs less time in a single iteration as in the following table. However, as we mentioned in the response, contrastive approaches require a **large batch size of at least 512**, to obtain current performance thus requires **large memory** to train the model. Our approach, on the other hand, could obtain similar performance with even less batch.
> > > >
> > > > Since uploading a revised version of the paper is not available now, we will leave the revised introduction and experimental details in the following response. Further, we will revise our manuscript as you suggested. We believe that incorporating your comments, as well as the comments from other reviewers, will significantly strengthen our paper.
> > > >
> > > >
> > > > Introduction Paragraph2
> > > > > However, both of these adversarial SSL frameworks are inefficient as they require a large batch size in order to attain good performances either on clean or adversarial samples [1,2]. When the memory and computation budgets are limited, such as when learning on resource-constrained devices, contrastive SSL may obtain poor performance due to the use of small batch sizes.
> > > >
> > > > > To be batch-size free SSL, recent SSL frameworks [3,4,5] mostly resort to maximizing the consistency across two differently augmented samples of the same instance, using an additional momentum encoder, without any negative pairs or additional networks.
> > > >
> > > > [1] Chen et al., A Simple Framework for Contrastive Learning of Visual Representations, ICML 2020 \
> > > > [2] Kim et al. Adversarial self-supervised contrastive learning., NeurIPS 2020 \
> > > > [3] Grill et al. Bootstrap your own latent: A new approach to self-supervised Learning, arXiv 2020 \
> > > > [4] Chen et al. Exploring Simple Siamese Representation Learning, CVPR 2021 \
> > > > [5] Zbontar et al. Barlow Twins: Self-Supervised Learning via Redundancy Reduction, ICML 2021
> > > >
> > > > Experiment Section 4.1. Paragraph 2
> > > > > As shown in the table, contrastive-based approaches **require twice more memory to obtain current performance** compared to positive-only SSL approaches. This further demonstrates and justifies why we need to achieve robustness with non-contrastive SSL methods, since non-contrastive methods not only significantly outperform contrastive methods, but requires less memory, which may be a clear practical advantage in a resource-constrained settings.
> > > >
> > > > | Model | Robustness | Time (m/epoch) | Memory (B) |
> > > > |------------------|:----------:|:--------------:|:----------:|
> > > > | RoCL | 42.89 | **6.86m** | 10.3G |
> > > > | ACL | 39.37 |11.92m |10.3G |
> > > > | TAROSS | 44.71 | 23.29m | **6.25G** |

---

> > > > > ### Comment · Reviewer_yLsz · 2022-11-26
> > > > > **Thanks for the update**
> > > > >
> > > > > Thanks for your update.
> > > > >
> > > > > The revised paragraphs and the new experiment looks good to me, so I raised my score from 3 to 6.
> > > > >
> > > > > Some other comments:
> > > > >
> > > > > [1] For the design of the score function, its design looks reasonable to me. I see from the authors' response to other reviewers about the ablation study. It would be helpful if there are more settings, e.g., different combinations of entropy and similarity.
> > > > >
> > > > > [2] Based on the new table, the speed of TAROSS is slower than RoCL. Could you provide some intuition on why the proposed algorithm is slower? Is it caused by the implementation?

---

> > > > > > ### Author Response · Authors · 2022-11-27
> > > > > > **Thanks for constructive comments**
> > > > > >
> > > > > > Thank you for your constructive comments.
> > > > > >
> > > > > > [1] Thank you for valuable comments. We will proceed with the ablation experiments on a different combination of each component. It will further strengthen our work.
> > > > > >
> > > > > > [2] It was caused by our code implementation. We attack different views of two images independently, so it takes twice more than ACL. However, we could shorten the time when we implement our algorithm to attack both images at the same time in a single batch as ACL does.
> > > > > >
> > > > > > Thanks again for your time and effort to review our paper.
> > > > > >
> > > > > > Thanks, \
> > > > > > Author

---

### Official Review · Reviewer_MNdh · 2022-10-22

**Confidence:** 4
**Correctness:** 3
**Technical Novelty And Significance:** 3
**Empirical Novelty And Significance:** 3
**Recommendation:** 6

**Clarity, Quality, Novelty And Reproducibility:**

Clarity: The manuscript is simple to read and follow.
Quality: The paper is interesting with strong motivation. However, some clarification and experiments are needed.
Novelty: The paper is novel from my best understanding.
Reproducibility: The presentation contains many details for reproducing the work.


**Strength And Weaknesses:**

Strength
(1).	 The motivation of this work is strong.
(2).	 Figure 1 clearly illustrates the difference between creating adversarial examples under the supervised setting and unsupervised setting. More specifically, the difference between creating adversarial examples in contrastive based self-supervised learning and positive only self-supervised learning is highlighted.
(3).	 The overall paper is clear and simple to follow.


Weaknesses
(1).	While the proposed method is useful under the scenario where only positive data is used for self-supervised learning, it is unclear why it cannot be applied to contrastive based self-supervised learning (e.g. SimCLR). If applicable to contrastive based self-supervised learning, how does the proposed method compared to prior works (e.g. RoCL)?
(2).	The author is suggested to explain why CIFAR5 is used in Table 1, instead of Cifar10. Moreover, how does table 1 in this paper compares to the Table 1 in RoCL? If the author is using Cifar 10, are Table 1 in this work comparable to Table 1 in RoCL?
(3).	Typically, self-supervised learning assumes that the pretraining feature can be applied to various downstream task. However, for the experiment in Figure3, the author assumes that the downstream task is known and use it to ``probe” and select the optimal range of the similarity score. The author is suggested to explain why such operation make sense in general self-supervised learning scenario. Will the selected similarity score range transfer across dataset? How does the similarity score selection affect the final performance?
(4).	It is unclear why the score function looks like Eq 6. Can we use the cosine similarity or the entropy term only? Which term is more important for the training? Any ablation on the weighted version between two terms?
(5).	Table 2 of RoCL also conducts experiment on black box attacks on Cifar10. What is the proposed method on black box attacks?


Typo:
(1).	“becase” : typo near table 2


**Summary Of The Paper:**

In this paper, the problem of adversarial self-supervised learning is studied, where the goal is to improve the robustness of the self-supervised model. While adversarial example can be created under the supervised scenario, creating adversarial example is not trivial for self-supervised training because no label is available. This paper addresses this problem by proposing a score function that measure the similarities between unlabeled samples. Given an unlabeled sample, the negative sample can then be selected and then used to guide the direction of the adversarial perturbation. Experiment demonstrates the effectiveness of the proposed method.

**Summary Of The Review:**

The author is suggested to address the limitation in the weakness section. It seems that some of the settings in this work are different than the prior works and several ablations are missing. The score might be adjusted if these concerns are not well addressed.

---

> ### Author Response · Authors · 2022-11-12
> **Response to comments (2/2)**
>
> **3. [Why use similarity score? how does the similarity score selection affect the final performance?]**
> >Typically, self-supervised learning assumes that the pretraining feature can be applied to various downstream task. However, for the experiment in Figure3, the author assumes that the downstream task is known and use it to ``probe” and select the optimal range of the similarity score. The author is suggested to explain why such operation make sense in general self-supervised learning scenario. Will the selected similarity score range transfer across dataset? How does the similarity score selection affect the final performance?
> - We **use a similarity score for empirical observation in adversarial self-supervised learning to explore intuition**. The results in Figure 3 shows the final score based on the range of similarity score selection.
> - First, please note that **we did not assume to know the downstream task and did not use it to probe the optimal range of similarity score.** Figure 3 is the results of observation to get an intuition of what will be the effective target in positive-only self-supervised learning.
> Self-supervised learning assumes that the pretraining feature can be applied to various downstream tasks. To do so, self-supervised learning has the goal to learn good visual representation which will eventually help in diverse tasks. To evaluate whether the quality of visual representation is well trained or not, linear evaluation is one of the metrics to verify the quality of representation.
> - **The results in Figure 3 are empirical observations and evaluate the visual representation.** Through this observation, interestingly we found the most far instances are bad choices, rather we should select adequately far instances as a target. Therefore, under this observation, we design the score function in Eq.6.
>
> **4. [Why can we just use only similarity term or entropy term?]**
> > It is unclear why the score function looks like Eq 6. Can we use the cosine similarity or the entropy term only? Which term is more important for the training? Any ablation on the weighted version between two terms?
> - Based on the ablation experiment, **both terms are essential to have better robustness** (following Table). Thanks to your comments, we will add the ablation experiment in the manuscript (Supplementary E.3.).
> - To explain why both terms are used, we will recap our observation in Section3.
> - First, we **empirically verify targeted attacks helps even when we select the target with the random sampling** in Table 1. Surprisingly, with the random sampling, relatively poor models of positive-only adversarial self-supervised learning framework shows improved clean and robust accuracy.
> - Then, based on our naive intuition and intuition from related work [Zhang et al., Chen et al.], we assume that confused samples or far away examples could help to generate effective targeted adversaries. Based on the observation in Table 2 and Figure 3, **selecting the confused and adequately far similar instance is effective in generating adversaries with targeted self-supervised attacks.** (Section 3.3, Paragraph. Entropy and similarity based target selection)
> - Therefore, we design the score function to have a composition of similarity score and entropy (it is not a cross entropy loss) which is based on our observation in the previous section. We could combine k-means clustering and similarity score, but to reduce the computation time of k-means clustering in every iteration, we naturally come up with entropy that could function similarly as a confused pseudo class which provides alternative class information without any class label.
>
> |                 | Clean     |    PGD    | AutoAttack |
> |:---------------:|-----------|:---------:|------------|
> |   Entropy term only  |   **78.43**      |  40.35     |   32.51     |
> | Similarity term only |  72.90   |  44.59   |  36.12   |
> |       Ours      | 74.06 | **44.71** | **36.39**  |
>
> [Zhang et al.] Attacks Which Do Not Kill Training Make Adversarial Learning Stronger, ICML 2020 \
> [Chen et al.] Improving Adversarial Robustness via Guided Complement Entropy, ICCV 2019
>
> **5. [Experiment on black box attack]**
> >Table 2 of RoCL also conducts experiment on black box attacks on Cifar10. What is the proposed method on black box attacks?
> - Thanks to your comments, we also evaluate our methods against black box attacks. Models on the row are the test models and models on the column are the source models. As shown in following tables, our model is more robust to AT model black box attack compare to RoCL.
>  |      | AT | RoCL | Ours |
> |:----:|----|------|-----|
> | AT   | -   |   59.73 | **60.92**   |
> | RoCL | 70.40   |  -    |  57.98   |
> | Ours |    69.97   | 54.99 | -   |

---

> > ### Comment · Reviewer_MNdh · 2022-11-22
> > **Thanks for your response**
> >
> > The rebuttal and the review from other reviewers have been read. The author has addressed most of my concerns in the rebuttal. However, the author also agreed that the selection of the score function is based on the empirical observation. This makes the proposed score function lack of theoretical insight, which is a concern also raised by other reviewers. I would like to maintain the initial score after the rebuttal.

---

> > > ### Author Response · Authors · 2022-11-25
> > > **Thank you for your valuable comments**
> > >
> > > Thank you for your response.
> > >
> > > We understand your concern that our score function is not based on theoretical ground. However, we believe that we have clearly shown the motivation and intuition regarding our score function with extensive empirical studies. Moreover, we also analyze why our score function works well on positive-only SSL in Section 4.3. Many existing works on adversarial robustness and self supervised learning provide useful insights and practically effective methods although they come without theoretical analysis [1-5], and we believe that ours is in line with those works, as we provided a very novel concept of "targeted attack" for adversarial self-supervised learning, and an effective method to select a target instance that works.
> > >
> > > Best,
> > > Author
> > >
> > > [1] Rubuffi et al., Data Augmentation Can Improve Robustness, NeurIPS 2021 \
> > > [2] Wu et al., Adversarial Weight Perturbation Helps Robust Generalization, NeurIPS 2020 \
> > > [3] Gowal et al., Improving Robustness using Generated Data, NeurIPS 2021 \
> > > [4] Chen et al., A Simple Framework for Contrastive Learning of Visual Representations, ICML 2020 \
> > > [5] He et al., Masked Autoencoders Are Scalable Vision Learners \
> > > [6] Xie et al., SimMIM: a Simple Framework for Masked Image Modeling

---

> ### Author Response · Authors · 2022-11-12
> **Response to comments (1/2)**
>
> **1. [Limited only on non-constrastive SSL, under what scenario does non-contrastive SSL is used?]**
> >While the proposed method is useful under the scenario where only positive data is used for self-supervised learning, it is unclear why it cannot be applied to contrastive based self-supervised learning (e.g. SimCLR). If applicable to contrastive based self-supervised learning, how does the proposed method compare to prior works?
> - We believe **when the computation budgets are limited such as in the edge devices, batch-free non-contrastive SSL could be applicable** than contrastive SSL which needs large batch-size.
> - Furthermore, as shown in the following table, **our approach is also applicable to positive pairs in contrastive SSL.** Although a positive pair is (1/Batch size) in contrastive SSL loss, so our approach is seemingly marginal than when we applied in non-contrastive SSL.
> - Experimental details for contrastive SSL with TAROSS is in the supplementary section E.
>
> |   Linear evaluation          | Clean     |    PGD     |
> |-----------|:-----------:|:---------:|
> | RoCL | 78.14     |   42.89   |
> | RoCL **+TAROSS** |  **78.43**   | **43.91**    |
> | ACL | 79.96 | 39.37 |
> | ACL **+TAROSS** | **80.02**| **40.18** |
>
> |   Robust linear evaluation         | Clean     |    PGD     |
> |-----------|:-----------:|:---------:|
> | RoCL | 76.53 | **47.51**  |
> | RoCL **+TAROSS** |  **80.11** | 46.47  |
> | ACL | 77.17 | 40.67 |
> | ACL **+TAROSS** |**79.84** | **42.29** |
>
> - The self-supervised framework has been developed **recently that does not limit the batch size and is able to obtain good performance with only positive pairs [2,3,4]** while vulnerability has never been considered.
> - We believe our first observation that a simple combination of adversarial learning and recent positive-only self-supervised could not have enough robustness is a surprising and important observation. Furthermore, we adequately demonstrate our intuition and the methods to overcome such vulnerability.
> - We believe that not only for state-of-the-art performance, but also for **discovering under explored problem and suggesting appropriate solutions based on the problem is also meaningful** in this community. Furthermore, we believe that **reverting recent SSL frameworks to contrastive learning does not seem a genuine solution** to ensuring the robustness of non-contrastive self-supervised learning frameworks.
> - We understand our approach could not show superior robustness in every task compared to contrastive-based learning, but **we resolve the vulnerability problem that non-contrastive SSL has where our work makes them have sometimes better robustness** than contrastive-based approaches.
> - In table 1, and 3, 4, our work with non-contrastive SSL shows **better robustness against PGD attack and strong AutoAttack** than contrastive SSL. Moreover, as we expected, our approach with non-contrastive SSL demonstrates **better transferred robustness which indicates our model benefits the high visual quality** of non-contrastive SSL while contrastive SSL could not.
>
> [1] Gowal et al., Self- supervised adversarial robustness for the low-label, high-data regime, ICLR 2021 \
> [2]Grill et al. Bootstrap your own latent: A new approach to self-supervised Learning, arXiv 2020  \
> [3] Chen et al. Exploring Simple Siamese Representation Learning, CVPR 2021 \
>  [4] Zbontar et al. Barlow Twins: Self-Supervised Learning via Redundancy Reduction, ICML 2021\
>
>
> **2. [Why table 1 is performed under CIFAR5?]**
> > The author is suggested to explain why CIFAR5 is used in Table 1, instead of Cifar10. Moreover, how does table 1 in this paper compares to the Table 1 in RoCL? If the author is using Cifar 10, are Table 1 in this work comparable to Table 1 in RoCL?
> - **We trained all models under CIFAR 5 in Table 1 for initial observation.** Due to high computation resources which takes about two days to train previous work and takes 4 days to train k-means clustering approach in Table 1 and Table 2, we downsize the dataset to CIFAR 5 in Table 1 and Table 2 to quickly verify our assumption that targeted attack could highly boost the positive-only self-supervised learning framework.
> - We clarify why we use CIFAR 5 in Table 1 and Table 2 in manuscript (blue text). Further, we also clarify in the caption which dataset is used in each table (blue text).

---

### Official Review · Reviewer_y32g · 2022-10-24

**Confidence:** 3
**Correctness:** 3
**Technical Novelty And Significance:** 2
**Empirical Novelty And Significance:** 2
**Recommendation:** 3

**Clarity, Quality, Novelty And Reproducibility:**

Overall the clarity is good and the quality is fair. The idea of the paper is easy to follow, the quality is affected by the factors I mentioned above.


**Strength And Weaknesses:**

Overall the idea has decent novelty and the overall idea of this paper is easy to grasp. What I like about this paper are:
1. The motivation is clearly stated: the untargeted attack fails to generate useful adversarial training data for improving robustness.
2. The experimental results are clearly in line of the arguments.

The limitations are also quite clear though, including:
1. The design of the score function (combing similarity and cross entropy) lacks justification. Although authors mentioned that we want to maximize the similarity but still unnatural to me why.
2. Some details are lacking. For instance the authors mentioned K-means multiple times, but based on what distance metric? How does this affect the training efficiency?
3. The title seems too broad - this paper concerns about SSL using positive pairs (specifically BYOL and Simsiam), but SSL itself is a very broad term containing many other methodologies as well. So the title might confuse others.
4. The comparison in the experiments are also concerning: the baseline is based on SimCLR (Jiang, Kim) but this paper is based on BYOL and Simsiam. Could this be the reason causing the performance improve? Authors might want to find more support.
5. The datasets are a bit small and limited, authors might want to find more realistic datasets to verify.


**Summary Of The Paper:**

The authors proposed a new unsupervised adversarial training method. Unlike previous work, this method leveraged label information and thus be categorized as the targeted attack. The core idea is the target selection algorithm, which is a combination of similarity and entropy. By running adversarial training on such perturbations, the authors show that this method enhances robustness of models trained with semi-supervised learning.

**Summary Of The Review:**

Please address some of my points above.

---

> ### Author Response · Authors · 2022-11-12
> **Response to comments (2/2)**
>
> **3. [Title is too broad]**
> > The title seems too broad - this paper concerns about SSL using positive pairs (specifically BYOL and Simsiam), but SSL itself is a very broad term containing many other methodologies as well. So the title might confuse others.
> - Thank you for your comments. We will **revise the title as follows to precisely describe our work more effectively.** “Targeted Adversarial Self-Supervised Learning using Positive Pairs“
> - Furthermore, **our approach is also applicable to positive pairs in contrastive learning methods** as follows. Thanks to the comments provided from Reviewer ngAb, MNdh, yLsz, we found that our approach also could improve the contrastive learning-based adversarial training.
>
> |   Linear evaluation          | Clean     |    PGD     |
> |-----------|:-----------:|:---------:|
> | RoCL | 78.14     |   42.89   |
> | RoCL **+TAROSS** |  **78.43**   | **43.91**    |
> | ACL | 79.96 | 39.37 |
> | ACL **+TAROSS** | **80.02**| **40.18** |
>
> |   Robust linear evaluation         | Clean     |    PGD     |
> |-----------|:-----------:|:---------:|
> | RoCL | 76.53 | **47.51**  |
> | RoCL **+TAROSS** |  **80.11** | 46.47  |
> | ACL | 77.17 | 40.67 |
> | ACL **+TAROSS** |**79.84** | **42.29** |
>
> **4. [Comparison seems wrong]**
> > The comparisons in the experiments are also concerning: the baseline is based on SimCLR (Jiang, Kim) but this paper is based on BYOL and Simsiam. Could this be the reason causing the performance improve? Authors might want to find more support.
> - The main comparison with our work is [Gowal et al.] which is untargeted adversarial self-supervised learning based on BYOL. Further, we also compared our approach with contrastive-based adversarial self-supervised learning based on SimCLR.
> - **Our performance gain is not from a self-supervised learning framework (BYOL or SimSiam).** Because, in Table 1 in Section 3.1, we observe **untargeted adversarial training based on BYOL and SimSiam could not show better robustness than SimCLR based approaches** (Jiang et al., Kim et al.).
> - To overcome such vulnerability in untargeted adversarial positive-only self-supervised learning, we propose a targeted attack based on our novel score function in Table 3, where we show improved robustness.
> - Based on our observation and analysis in Figures 4, 5, our approach selects the target that is confused and effective samples to generate effective adversaries with only positive pairs. Therefore, effective adversaries make BYOL and SimSiam have improved robustness that untargeted BYOL and SimSiam could not have.
>
> **5. [Realistic datasets should be verify]**
> >The datasets are a bit small and limited, authors might want to find more realistic datasets to verify.
> - Please note that **we verify our work on CIFAR10, CIFAR100, and STL10** which we followed previous adversarial self-supervised learning approaches [Jiang et al., Kim et al.] that are standard benchmarks in adversarial self-supervised learning.
>
> [Gowal et al.] Self- supervised adversarial robustness for the low-label, high-data regime, ICLR 2021 \
> [Jian et al.] Robust pre-training by adversarial contrastive learning. , NeurIPS 2020 \
> [Kim et al.] Adversarial self-supervised contrastive learning., NeurIPS 2020

---

> ### Author Response · Authors · 2022-11-12
> **Response to comments (1/2)**
>
> **1. [Score function lacks justification]**
> >The design of the score function (combing similarity and cross entropy) lacks justification. Although authors mentioned that we want to maximize the similarity but still unnatural to me why.
> - **We design our score function based on the observation in Section 3**. Therefore, we will recap our observations.
> - First, we **empirically verify targeted attacks helps** even when we select the target with the random sampling in Table 1. Surprisingly, with the random sampling, relatively poor models of positive-only adversarial self-supervised learning framework shows improved clean and robust accuracy.
> - Then, based on our naive intuition and intuition form previous work [Zhang et al., Chen et al], we assume that confused samples or far away examples could help to generate effective targeted adversaries. Based on the observation in Table 2 and Figure 3, **selecting the confused and adequately similar instance is effective in generating adversaries using targeted self-supervised attacks.** (Section 3.3, Paragraph. Entropy and similarity based target selection)
> - Therefore, we design **the score function to have composition of similarity score and entropy (it is not a cross entropy loss) which is based on our observation** in the previous section. We could combine k-means clustering and similarity score, but to reduce the computation time of k-means clustering in every iteration, we naturally come up with entropy that could function similarly as a confused pseudo class which provides alternative class information without any class label.
> - Furthermore, we also experimented ablation study of our score function .
>
> |                 | Clean     |    PGD    | AutoAttack |
> |:---------------:|-----------|:---------:|------------|
> |   Entropy term only  |   **78.43**      |  40.35     |   32.51     |
> | Similarity term only |  72.90   |  44.59   |  36.12   |
> |       Ours      | 74.06 | **44.71** | **36.39**  |
>
> [Zhang et al.] Attacks Which Do Not Kill Training Make Adversarial Learning Stronger, ICML 2020 \
> [Chen et al.] Improving Adversarial Robustness via Guided Complement Entropy, ICCV 2019
>
> **2. [Experimental details are missing]**
> > Some details are lacking. For instance the authors mentioned K-means multiple times, but based on what distance metric? How does this affect training efficiency?
> - Thank you for your suggestion. We clarify the experimental details in the paper (blue text).
> - We use K-means clustering with the metric of cosine similarity distance. To conduct the K-means clustering at every interaction, it takes 2.2 times more than our TAROSS to train. Thanks to your comments, we also revised the main texts.

---

> ### Author Response · Authors · 2022-11-25
> **Gentle reminders**
>
> Dear reviewer y32g,
>
> Thank you for your time and effort to review our paper.
> During the rebuttal, we did our best to resolve your initial concerns. Could you go over our rebuttal and check our response?
>
> - We demonstrate **additional experiments that applied our TAROSS on contrastive self-supervised learning.** Our targeted attack not only improves robustness of non-contrastive SSL, but also contrastive SSL.
>
> - Furthermore, we **explain how our approach is designed based on empirical observation.** We **additionally demonstrate each component in the score function experimentally.**
>
> - We also revised our manuscripts based on your technical comments and missing experimental details.
>
> Therefore, we believe we did our best to resolve your concerns. Therefore, we hope you find our response and reflect this on your final review and the score. Please kindly let us know if there are any further questions or concerns. We would be more than happy to address them.
>
> We thank you again for your time and efforts in reviewing our paper.
>
> Best, \
> Authors

---

> ### Author Response · Authors · 2022-11-27
> **A Gentle Reminder**
>
> Dear reviewer y32g,
>
> Thank you again for your time and efforts in reviewing our paper. This is a gentle reminder regarding our response, since we have not yet received feedback on it. We believe that we have faithfully addressed all your concerns in our responses we have previously uploaded. We summarize our responses for your convenience.
>
> Your initial concerns were as follows:
> 1. The design of the score function (combing similarity and cross entropy) lacks justification. Although authors mentioned that we want to maximize the similarity but still unnatural to me why.
> 	> We design the score function based on the observation in Section 3. In section 3, we find that we need to find sufficiently far from the original instance, while not trivially distinguishable and remains confusing, as a target of instance-wise attack. **The empirical analysis in the section suggests that when we select a confusing and sufficiently far target instance, it can become a stronger adversarial example that are near the boundaries of each cluster, which is effective for adversarial training without any label information.**
>
> 2. Some details are lacking. For instance the authors mentioned K-means multiple times, but based on what distance metric? How does this affect the training efficiency?
> 	> We revised the manuscripts to address these points (distance metric and its effect on training efficiency).
>
> 3. The title seems too broad - this paper concerns about SSL using positive pairs (specifically BYOL and Simsiam), but SSL itself is a very broad term containing many other methodologies as well. So the title might confuse others.
> 	> Since **our method is also applicable to contrastive-based SSL**  such as SimCLR, and we have included results on contrastive SSL in the revised version of the paper, we believe that we could revise our title as “Targeted Instance-Wise Attacks for Adversarial Self-Supervised Learning".
>
> 4. The comparison in the experiments are also concerning: the baseline is based on SimCLR (Jiang, Kim) but this paper is based on BYOL and Simsiam. Could this be the reason causing the performance improve? Authors might want to find more support.
> 	> This is a misunderstanding of our work because **our work is motivated by the observation that NC-SSL methods such as BYOL and SimSiam do not achieve good robust accuracy when we apply untargeted attacks as we do for the C-SSL frameworks**, compared to robust C-SSL frameworks such as ACL [Jiang et al.] and RoCL [Kim et al.] (Section 3).
>
> Thus, use of BYOL and SimSiam as baselines are definitely not the reason our method improves, as can be seen from their poor performance with untargeted attacks. Rather, the improvement is coming from the effectiveness of the targeted attack we propose for the  instance-wise attacks.
>
> 5. The datasets are a bit small and limited, authors might want to find more realistic datasets to verify.
> 	> Please note that **we experiment on standard benchmark datasets for adversarial SSL, following the experimental settings of the previous works [Jiang et al., Kim et al.], and adversarial robustness in general**. We agree that the datasets are smaller compared to the ones that are used for standard SSL, but this is the current standard and we need to follow them to compare against reported results.
>
> More detailed responses for each comment could be found in the following threads. We hope that this explanation resolves your concerns on our work. Please let us know if there are any remaining concerns since we will be more than happy to address them.
>
> Thanks, \
> Authors

---

> ### Author Response · Authors · 2022-12-02
> **A Gentle Reminder**
>
> Dear y32g,
>
> Thank you again for your time and efforts in reviewing our paper. In the author responses below, we have faithfully addressed all your comments, answering your questions, correcting factual misunderstandings, and providing additional experiments to address your points. We hope they resolve your concerns, and please let us know if there is anything else we need to address.
>
> Best,\
> Authors

---

### Official Review · Reviewer_ngAb · 2022-10-24

**Confidence:** 4
**Correctness:** 3
**Technical Novelty And Significance:** 2
**Empirical Novelty And Significance:** 2
**Recommendation:** 6

**Clarity, Quality, Novelty And Reproducibility:**

Writing and clarity are not the strong points of the paper. The notation is confusing and often wrong. The text misses many important details that hinder the main authors' findings.

Technical comments:
- to obtain significantly high robustness -> to obtain significantly *higher* robustness
- "SimSiam detaches the gradient on the z" - it is better to say that it stops gradients for z-part
- p_2 and z_1 are not defined
- in formula (6) indexes should be inside the brackets $x)_i$ -> $x_i)$
- I also suppose that indexes for T_i are not correct in the same formula and around it
- Table 3 - what is the used metric?
- I think it is unacceptable to use bold font in Table to articulate the results of your approach, as the bold font is often used to show the best results, which is not the case in most experiments
- Visualization of embeddings space: I suppose that the authors use tSNE but never mention it

**Strength And Weaknesses:**

Strengths:
- The problem the authors propose is interesting and lacks a straightforward solution
- The authors propose an interesting idea on how to incorporate probably-negative objects in the framework

Weaknesses:
- The authors only partially solve the adversarial training for NC-CCL. C-SSL methods work better according to my interpretation of the presented results
- There is no principal explanation of the usage of the approach at hand. There is some experimental analysis, but, e.g. it is missing if we can achieve similar results with different types of losses inspired by similar ideas.

Additional comments:
The idea of selecting a far-away image as a starting point resemble many black-box adversarial attacks.

**Summary Of The Paper:**

The paper discusses the problem of adversarial training for non-contrastive self-supervised learning (NC-SSL) approaches like BYOL.
They propose a modification of the loss function that involves an adversarial attack for a pair of objects that, with high probability, lie in different classes.
Such a method allows to beat straightforward adversarial training for NC-SSL, but is not Pareto-optimal given a clean and robust accuracy compared to C-SSL methods like SimCLR.

**Summary Of The Review:**

The idea is good. The method is not good enough from theoretical and practical points of view, with little novelty in the current approach.
I also note that may be other approaches applied to robustness improvement can work in this case, like SAM or Differential Privacy algorithms with a proper algorithm.
I suggest authors provide a deeper analysis of why they obtained these results and why NC-SSL fails to provide robust solutions.

---

> ### Author Response · Authors · 2022-11-12
> **Response to your comments (2/2)**
>
> **[Weakness] The method is not good enough from theoretical and practical points of view, with little novelty in the current approach**
> - Our paper is not a theoretical paper but **we provide extensive empirical observation and results.** We believe empirical exploration is also meaningful in the academic community.
> - Further, our approach is **applicable to any type of self-supervised learning framework (e.g. SimCLR, BYOL, and SimSiam)** which we believe is a high practical contribution.
> - Moreover, our work first explores the limitation that a positive-only self-supervised learning framework is vulnerable. To overcome such vulnerability, we propose a targeted attack which has never been introduced in adversarial self-supervised learning. Especially, targeted attacks based on our novel score function shows improved performance in robustness of SSL.
> - Therefore, we believe our approach demonstrates various novelties that could bring benefits to the community, so please consider these novel points of our work that you might have missed.
>
> **[Technical comments]**
> - Thank you for your technical comments. We revise the main manuscript based on your comments. (blue texts)
> - The metric of Table 3 is classification accuracy.
> - We use tSNE to visualize Figure 5.

---

> > ### Comment · Reviewer_ngAb · 2022-12-01
> > **Score raised a bit given the comments**
> >
> > Dear all! I thank the authors for the additional results and discussions they presented during the rebuttal. I raise my score to slightly above the acceptance threshold given the comments above. I hope, that this paper will lead to the further development of the proposed problem in the field.

---

> > > ### Author Response · Authors · 2022-12-01
> > > **Thank you for your time to review our paper.**
> > >
> > > Dear Reviewer ngAb,
> > >
> > > Thank you for your efforts in reviewing our paper.
> > > We are happy that we resolved your concerns during the rebuttal.
> > > Moreover, we sincerely thank again for your time and your valuable comments to further strengthen our paper.
> > >
> > > Best, \
> > > Authors

---

> ### Author Response · Authors · 2022-11-12
> **Response to your comments (1/2)**
>
> **1. [Limited only on non-constrastive SSL]**
> > The authors only partially solve the adversarial training for NC-CCL. C-SSL methods work better according to my interpretation of the presented results
> - Thanks for your constructive comments. As shown in the following table, **our approach is also applicable to positive pairs in C-SSL**. Since a positive pair contributes (1/Batch size) to total C-SSL loss, gain is seemingly more marginal than when we applied TAROSS in NC-SSL.
> - Experimental details for NC-SSL with TAROSS are in **supplementary section E.**
>
> |   Linear evaluation          | Clean     |    PGD     |
> |-----------|:-----------:|:---------:|
> | RoCL | 78.14     |   42.89   |
> | RoCL **+TAROSS** |  **78.43**   | **43.91**    |
> | ACL | 79.96 | 39.37 |
> | ACL **+TAROSS** | **80.02**| **40.18** |
>
> |   Robust linear evaluation         | Clean     |    PGD     |
> |-----------|:-----------:|:---------:|
> | RoCL | 76.53 | **47.51**  |
> | RoCL **+TAROSS** |  **80.11** | 46.47  |
> | ACL | 77.17 | 40.67 |
> | ACL **+TAROSS** |**79.84** | **42.29** |
> - The self-supervised framework has been developed **recently that does not limit the batch size and is able to obtain good performance with only positive pairs [2,3,4]** while vulnerability has never been considered.
> - We believe our first observation that a simple combination of adversarial learning and recent positive-only self-supervised could not have enough robustness is a surprising and important observation. Furthermore, we adequately demonstrate our intuition and the methods to overcome such vulnerability.
> - We believe that not only for state-of-the-art performance, but also for **discovering under explored problem and suggesting appropriate solutions based on the problem is also meaningful** in this community. Furthermore, we believe that **reverting recent SSL frameworks to contrastive learning does not seem a genuine solution** to ensuring the robustness of non-contrastive self-supervised learning frameworks.
> - We understand our approach could not show superior robustness in every task compared to contrastive SSL, but **we resolve the vulnerability problem that non-contrastive SSL has where our work makes them have sometimes better robustness** than contrastive-based approaches. Moreover, **during the rebuttal, we found that our approach also improves robustness of previous adversarial C-SSL.**
> - In table 1, and 3, 4, our work with non-contrastive SSL shows **better robustness against PGD attack and strong AutoAttack** than contrastive SSL. Moreover, as we expected, our approach with non-contrastive SSL demonstrates **better transferred robustness which indicates our model benefits the high visual quality** of non-contrastive SSL while contrastive SSL could not.
>
> [1] Gowal et al., Self- supervised adversarial robustness for the low-label, high-data regime, ICLR 2021 \
> [2] Grill et al. Bootstrap your own latent: A new approach to self-supervised Learning, arXiv 2020  \
> [3] Chen et al. Exploring Simple Siamese Representation Learning, CVPR 2021 \
>  [4] Zbontar et al. Barlow Twins: Self-Supervised Learning via Redundancy Reduction, ICML 2021
>
> **2. [No principal explanation of the usage of the approach]**
> >There is no principal explanation of the usage of the approach at hand. There is some experimental analysis, but, e.g. it is missing if we can achieve similar results with different types of losses inspired by similar ideas.
> - Our approach is **based on our empirical observation in Section 3**. We extensively demonstrate how targeted attack affects positive-only self-supervised learning.
> - First, we **explore a simple random selection** of targets in targeted attack in Table 1. It could boost both clean and adversarial performance in positive-only self-supervised learning.
> - Further, we **empirically observe how to select the target** to acquire better robustness based on the diverse similarity score range, and k-means clustering approaches in Section 3.2.
> - Finally, based on the observation, we conclude that selecting adequately far but selecting confusing class as a target instance for targeted attack helps in positive-only self-supervised learning in section 3.3. Therefore, **we propose our novel score function consists of similarity score and entropy score**. Moreover, we verify the consistent effectiveness of our approach in different types of NC-SSL frameworks, e.g., BYOL and SimSiam in Table 3 (summarized in the following table).
> - Therefore, we believe **our approach has extensive empirical explanation**.
>
> |        | Attack type  | Clean     |    PGD    |
> |--------|:----:|-----------|:---------:|
> | BYOL | untargeted | 72.65    | 16.20 |
> | BYOL | Ours | **84.52**  | **31.19** |
> | SimSiam | untargeted | 71.78    | 37.28 |
> | SimSiam | Ours | **74.06** | **44.71** |

---

> ### Author Response · Authors · 2022-11-25
> **Gentle reminders,**
>
> Dear reviewer ngAb,
>
> We resolved your two concerns 1) current work only partially solves the adversarial training for non-contrastive self-supervised learning and 2) no principal explanation of the usage of the approach.
>
> We demonstrate **additional experiments that applied our TAROSS on contrastive self-supervised learning.** Our targeted attack also improves adversarial contrastive SSL.
>
> Furthermore, we **explain how our approach is designed based on empirical observation.** Moreover, we **additionally demonstrate each component in the score function experimentally** (Please check on the responses to reviewer y32g).
>
> We believe we did our best to resolve your concerns. Therefore, we hope you find our response and reflect on your final review and the score. Please kindly let us know if there are any further questions or concerns. We would be more than happy to address them.
>
> We thank you again for your time and efforts in reviewing our paper.
>
> Best, \
> Authors

---

> ### Author Response · Authors · 2022-11-27
> **A Gentle Reminder**
>
> Dear reviewer ngAb,
>
> We would like to gently remind you about our response, since we have not yet received feedback on it.
>
> You raised two initial concerns: 1) our work only partially solves the adversarial training for NC-CCL, and 2) there is no principal explanation of the usage of the approach at hand.
>
> However, we believe we have successfully resolved these issues in our rebuttal. First of all, **our work is also applicable to both NC-SSL and C-SSL**. Although we have focused on improving the robust training with NC-SSL framework due to the popularity of NC-SSL methods and the ineffectiveness of untargeted attacks for NC-SSL, its target selection process can be applied to C-SSL framework (selecting the most confusing positive instance) and thus TAROSS is applicable to C-SSL to enhance its robust and clean accuracy.
>
> We additionally demonstrate a loss ablation experiment which shows that **our targeted attack could obtain consistently better robustness than an untargeted attack with different types of the score function**. However, our proposed score function is the most effective function to select the proper target. Furthermore, **our TAROSS consistently improves different NC-SSL frameworks**, i.e., BYOL and SimSiam, which utilize different objectives. Lastly, we believe that provided reasonable methods to overcome the vulnerability of NC-SSL based on the observation in Section 3 as reviewer yLsz points out.
>
> Please go over our responses and let us know if you have any remaining questions or concerns. We will be more than happy to address them.
>
> Best, \
> Authors

---

### Official Review · Reviewer_h3Xt · 2022-12-10

**Confidence:** 4
**Correctness:** 3
**Technical Novelty And Significance:** 2
**Empirical Novelty And Significance:** 2
**Recommendation:** 3

**Clarity, Quality, Novelty And Reproducibility:**

[Clarity is poor] As stated in Weakness, Authors do not provide the definitions of some terms and some important preliminaries.

[Quality is poor] As stated in Weakness, the effectiveness of the proposed method is not well validated. In addition, the target selection method is a bit tricky and lacks theoretical justification, which makes its quality further degrade.

[Novelty is marginal]: The paper replaces the label with a carefully-selected target data for generating targeted adversarial data in the setting of adversarial SSL, which is a marginal improvement in novelty.

[Reproducibility seems to be good.] The paper provides experimental details in the section of Reproducibility Statement.


**Strength And Weaknesses:**

Strength:
[Motivation for target attacks is clear.] Figure 1 clearly illustrates that targeted adversarial data can help find effective adversarial data in the setting of adversarial SSL with only positive pairs.

Weakness:
[Writing is poor.]
The definitions of some terms in the abstract and introduction like “positive-only adversarial SSL” are unclear to readers. In addition, Authors cannot assume the reader fully read “BYORL”. The paper does not introduce BYROL in detail. The paper should be self-contained. Besides, please write a full name before using the abbreviations.

[Motivation is unclear.]
This paper tries to propose a method to improve adversarial self-supervised learning. There already exists effective adversarial contrastive learning based on SimCLR. However, this paper tries to improve a suboptimal method BYORL.

[Effectiveness of the proposed method is not well validated.]
-In the introduction, it seems that Authors tried to improve non-contrastive methods since non-contrastive methods are more efficient. However, Authors do not show the efficiency of the proposed targeted SSL compared to BYORL. Observed that TAROSS spends almost twice of the training time of ACL from the results in Author’s response (https://openreview.net/forum?id=wwRjJScpsOO&noteId=BU4byVrzGsk), it seems that the proposed targeted SSL is more inefficiency.
-The scalability and efficiency of the targeted SSL on the large-scale datasets (e.g., ImageNet) are unknown. It would be more difficult to select an appropriate target sample in a large-scale dataset, which could degrade the efficiency and scalability of the proposed method.
-The clean accuracy of the proposed targeted SSL is significantly degraded on some datasets (such as STL10 and CIFAR100 in Table 4). The degradation of clean accuracy is even higher than the improvement in robust accuracy. Therefore, targeted SSL could be problematic.


**Summary Of The Paper:**

The paper aims to improve the robustness of adversarial self-supervised learning (SSL) by leveraging the proposed targeted adversarial data for adversarial SSL. The targeted adversarial data is generated by updating the natural data towards a targeted sample selected by K-mean or similarity. The paper empirically validates that target attacks can improve the adversarial robustness of several previous adversarial SSL methods including BYORL and SimSiam.

**Summary Of The Review:**

This paper proposes a target attack for improving the robustness of adversarial SSL. However, the paper’s weighting is poor, its motivation is unclear, and it lacks theoretical justification and enough empirical results to validate its effectiveness.

---

> ### Author Response · Authors · 2022-12-10
> **Response to your comment (3/3)**
>
>
> [Clarity is poor] As stated in Weakness, Authors do not provide the definitions of some terms and some important preliminaries.
> * We believe that we did clearly describe the important preliminaries and terminologies in the Related works and Methods sections. We will include additional descriptions of BYORL in the revision for better clarity.
>
> [Quality is poor] As stated in Weakness, the effectiveness of the proposed method is not well validated. In addition, the target selection method is a bit tricky and lacks theoretical justification, which makes its quality further degrade.
> * We have shown its effectiveness with diverse self-supervised learning frameworks (e.g., SimCLR, BYOL, and SimSiam) without extensive hyperparameter tuning. Thus we do not believe that our method is tricky but rather a simple yet effective method.
> * Our target selection method is **based on intuitive objectives and extensive empirical analysis** of which targeted samples can help improve non-contrastive SSL to generate more effective adversaries to obtain better robustness.
> * Moreover, we also analyzed why our score function works well on positive-only SSL in Section 4.3. It selects more confusing examples as the target.
> * Many existing works on adversarial robustness provide useful insights and practically effective methods although they come without theoretical analysis [1-3], and we believe that ours is in line with those works. In this paper, we proposed a novel concept of **targeted attack for adversarial self-supervised learning**, and an effective method to select a target instance that works.
>
> [Novelty is marginal]: The paper replaces the label with a carefully-selected target data for generating targeted adversarial data in the setting of adversarial SSL, which is a marginal improvement in novelty.
> * This is a critical misunderstanding. The idea of **generating targeted adversarial data in the setting of adversarial SSL has never been introduced** in adversarial self-supervised learning literatures before. Therefore, we believe the whole targeted adversarial attack idea is novel.
> * Furthermore, our work is the first to explore the limitation of untargeted attacks for adversarial self-supervised learning.
> * Therefore, we believe our approach is novel in three aspects: 1) the problem it targets, 2) findings from the empirical analysis, and 3) the methodology. We also believe that our introduction of targeted attack for adversarial self-supervised learning can lead to further research into the direction.
>
> [1] Rubuffi et al., Data Augmentation Can Improve Robustness, NeurIPS 2021 \
> [2] Wu et al., Adversarial Weight Perturbation Helps Robust Generalization, NeurIPS 2020 \
> [3] Gowal et al., Improving Robustness using Generated Data, NeurIPS 2021

---

> ### Author Response · Authors · 2022-12-10
> **Response to your comment (2/3)**
>
> [Effectiveness of the proposed method is not well validated.]
> > In the introduction, it seems that the Authors tried to improve non-contrastive methods since non-contrastive methods are more efficient. However, Authors do not show the efficiency of the proposed targeted SSL compared to BYORL. Observed that TAROSS spends almost twice of the training time of ACL from the results in Author’s response (https://openreview.net/forum?id=wwRjJScpsOO&noteId=BU4byVrzGsk), it seems that the proposed targeted SSL is more inefficiency.
> * First of all, the **non-contrastive learning approach is more efficient on computational budgets** since it **does not require a large batch size**. Therefore, our method is more efficient than contrastive adversarial SSL methods as far as we employ our approach on the non-contrastive self-supervised learning framework.
> * Moreover, as we commented on that response (https://openreview.net/forum?id=wwRjJScpsOO&noteId=WPZn_pTWpUv), **our approach results in similar training time as ACL** when we utilize single adversarial attacks with respect to two views. The initial difference was more of an implementation issue and we mentioned this in the later response.
> * Furthermore, BYORL utilized 40 steps of PGD attacks or even AutoAttacks during the training which require extremely large computational cost as we mentioned in Supplementary A. Therefore, **we could confidently claim that our approach is more efficient than BYORL**.
>
> > The scalability and efficiency of the targeted SSL on large scale datasets (e.g., ImageNet) are unknown. It would be more difficult to select an appropriate target sample in a large-scale dataset, which could degrade the efficiency and scalability of the proposed method.
> * Please note that **we verify our work on the exactly the same set of datasets** that were used in previous works on adversarial self-supervised learning [Jiang et al., Kim et al.]. They are simply standard benchmark datasets.
> * Moreover, we believe our work is more effective and scalable to large scale datasets as shown in CIFAR100, since when the number of class increases, the contrastive-based approach becomes more sensitive to the batch size since there is less number of instances of the same class in the single batch. However, our approach finds the most effective target regardless of the batch size.
>
> > The clean accuracy of the proposed targeted SSL is significantly degraded on some datasets (such as STL10 and CIFAR100 in Table 4). The degradation of clean accuracy is even higher than the improvement in robust accuracy. Therefore, targeted SSL could be problematic.
> * As shown in Table 4, our approach, **TAROSS, improves both the clean and robustness of positive-only self-supervised learning** compared to untargeted adversarial self-supervised learning.
> * Moreover, when **we applied our approach to contrastive-based adversarial self-supervised learning, TAROSS improves both clean and robustness** as shown in the following table.
>
> | Linear evaluation | Clean | PGD |
> |-----------|:-----------:|:---------:|
> | RoCL | 78.14 | 42.89 |
> | RoCL **+TAROSS** | **78.43** | **43.91** |
> | ACL | 79.96 | 39.37 |
> | ACL **+TAROSS** | **80.02**| **40.18** |
>
> | Robust linear evaluation | Clean | PGD |
> |-----------|:-----------:|:---------:|
> | RoCL | 76.53 | **47.51** |
> | RoCL **+TAROSS** | **80.11** | 46.47 |
> | ACL | 77.17 | 40.67 |
> | ACL **+TAROSS** |**79.84** | **42.29** |
>
> [Jiang et al.] Robust pre-training by adversarial contrastive learning., NeurIPS 2020  \
> [Kim et al.] Adversarial self-supervised contrastive learning., NeurIPS 2020

---

> ### Author Response · Authors · 2022-12-11
> **Response to your comments (1/3)**
>
> We sincerely thank you for your effort and additional time to review our paper.
> Thanks for your review even though it is not an official timeline to do the review of our work.
> We appreciate your detailed comments. Thus, we politely ask you to go over our response which might resolve your initial concerns.
>
>
> [Writing is poor.] The definitions of some terms in the abstract and introduction are unclear to readers.
> > The definitions of some terms in the abstract and introduction like “positive-only adversarial SSL” are unclear to readers. In addition, Authors cannot assume the reader fully read “BYORL”. The paper does not introduce BYROL in detail. The paper should be self-contained. Besides, please write a full name before using the abbreviations.
> * We could not contain a more detailed definition of positive-only SSL in the abstract. However, we will revise the introduction with a more detailed BYORL compared to previous contrastive-based adversarial SSL that could be more gentle to readers.
> * However, **we believe our paper is still self-contained about the description of BYORL**. We first simply describe the BYORL in the **introduction** and give a more **detailed algorithm in Section 2. Related work**. We also include the **experimental details in Supplementary A**.
>
>
> [Motivation is unclear.] This paper tries to propose a method to improve adversarial self-supervised learning. There already exists effective adversarial contrastive learning based on SimCLR. However, this paper tries to improve a suboptimal method of BYORL.
> * This is a misunderstanding of our motivation because as other reviewers commented, **we start from the critical and interesting problem that the recent self-supervised framework (which utilizes only positive pairs to calculate the loss) could not show enough robustness** as a contrastive-based framework. We observe that untargeted attacks with only positive pairs may generate ineffective adversaries compared to the adversaries that are generated with contrastive loss.
> * The reason why positive-only self-supervised learning frameworks should be improved is that those frameworks explicitly have merits compared to contrastive-based frameworks. **Positive-only self-supervised learning frameworks are not restricted to the batch size which will let us apply the model in the limited computational budget**, i.e., edge devices, while contrastive learning could not. Therefore, we believe it is worth overcoming such vulnerabilities.
> * Moreover, in the additional experiments we provided in our responses to the reviewers, we demonstrated that **our approach is also applicable to contrastive adversarial self-supervised learning frameworks** to improve their robust accuracy.
>
> | Linear evaluation | Clean | PGD |
> |-----------|:-----------:|:---------:|
> | RoCL | 78.14 | 42.89 |
> | RoCL **+TAROSS** | **78.43** | **43.91** |
> | ACL | 79.96 | 39.37 |
> | ACL **+TAROSS** | **80.02**| **40.18** |

---

### Author Response · Authors · 2022-11-18
**A summary of paper revision**

We really appreciate all the reviewers for their constructive comments. We have faithfully responded to the individual comments from the reviews. We believe we resolve most of reviewers' concerns during the rebuttal. We have included the results to resolve the major concerns in the revision. Here, we briefly summarize the updates we have made to the revision:

1. limited on the contrastive based approach
> **Main Section 4.1, Supplementary Section E.1.** We applied our approach on the contrastive based approach, e.g., RoCL, and ACL.
2. Ablation experiment on the score function
> **Supplementary Section E.2.** We conduct ablation experiment on the score function.
3. Robustness against black box attack
> **Supplementary Section E.3.** We conduct black box experiment.

---

### Decision · Program_Chairs · 2023-01-20

**Decision:**

Reject

**Justification For Why Not Higher Score:**

This paper provides some beneficial points to the community, e.g., the interesting problem setting and core idea. However, as mentioned in the mete review, this paper needs careful polishing to make a stronger submission. In the internal discussion, AC and reviewers reached a consensus that the current version of this paper cannot be accepted.

**Justification For Why Not Lower Score:**

N/A

**Metareview: Summary, Strengths And Weaknesses:**

**Summary**

This paper focuses on adversarial self-supervised learning. The aim is to improve the robustness of the self-supervised model. By proposing a score function that measures the similarities between unlabeled data, the targeted adversarial data can be generated. Experiments demonstrate that targeted attacks can improve the adversarial robustness of several self-supervised learning methods.

**Strengths**
- The motivation is strong. This paper clearly describes how the untargeted attack fails to generate useful adversarial training data for improving robustness.
- The proposed idea is interesting and can inspire follow-up research.
- The overall paper is clear. It is easy to follow technical details.

**Weaknesses**
- The method design lacks sufficient support and justifications, which makes it not very convincing to readers. Besides, the novelty compared with previous work is not well explained.
- Experimental verifications are insufficient in the present version. The claims of this paper should be further supported on large-scale datasets (e.g., ImageNet).

This paper achieves mixed recommendations from reviewers. The reviewers who are positive consider that this paper provides an interesting point to the research community. However, after rebuttal, all the reviewers still think this paper (a) lacks enough justifications for method designs, e.g., the selection of score functions; (b) the effectiveness of the proposed method is not well validated. (c) writing should be improved and becomes more logical. Overall, the current paper is not ready for publication. Authors could refer to the reviews and take suggestions to make a stronger submission. We believe the paper would be a strong one after addressing the above concerns.